# Unleashing Implicit Rewards:
# Prefix-Value Learning for Distribution-Level Optimization

**Shiping Gao** [1]   **Hongzhan Chen** [1]   **Xiaojun Quan** [1,2]   **Qifan Wang** [3]   **Lifu Huang** [4]

## Abstract

Process reward models (PRMs) provide fine-grained supervision for reasoning, but reliable PRMs often require step annotations or heavy verification pipelines, making them costly to scale and refresh during online RL. Implicit PRMs reduce this cost by training log-likelihood-ratio rewards from trajectory-level outcome labels. However, the log-ratio is constrained only as a sequence-level aggregate during training, while inference decomposes it into token- or step-level scores for partial prefixes. This train–inference mismatch leaves local credits weakly identified, so distribution-wide scoring can amplify misleading advantages. We propose *Implicit Prefix-Value Reward Model* (IPVRM), which directly learns the probability of eventual correctness for each prefix from outcome labels. Step signals are then obtained as temporal-difference (TD) differences between consecutive prefix values, aligning the training target with inference-time use. IPVRM markedly improves step-verification $F_1$ on PROCESSBENCH. To exploit these prefix values during policy optimization, we further introduce *Distribution-Level RL* (DistRL), which applies TD advantages to both sampled tokens and high-probability candidate tokens, providing dense counterfactual updates without additional rollouts. Experiments show that DistRL brings limited gains with unreliable implicit rewards, but consistently improves downstream reasoning when paired with IPVRM. The implementation of our method is available at https://github.com/gaoshiping/IPVRM.

## 1. Introduction

Large language models are increasingly trained with reinforcement learning from verifiable rewards (RLVR) (Lambert et al., 2025; Guo et al., 2025), where an automatic verifier provides a reliable terminal correctness signal. However, a single outcome label is shared by all tokens in a long reasoning trajectory, making it hard to identify which intermediate step caused success or failure; this weak credit assignment typically increases exploration and hurts sample efficiency. Process reward models (PRMs) (Lightman et al., 2023) aim to provide fine-grained supervision over the reasoning process, improving credit assignment beyond outcome-only rewards for RL training and supporting test-time inference. In practice, PRMs face a fidelity–cost trade-off: higher step-wise reliability typically requires more expensive data collection or heavier verification procedures. Explicit PRMs (Wang et al., 2024; Lu et al., 2024) rely on step-level supervision, while generative PRMs (Zhao et al., 2026) produce rationales or critiques before scoring, improving semantic interpretability but adding data or inference overhead. These costs are especially problematic in RL, where policy drift calls for frequent reward-model refreshes to mitigate distribution shift and reward hacking.

Implicit PRMs (Yuan et al., 2025a) reduce supervision cost by training log-likelihood-ratio rewards from sequence-level feedback, such as outcome verifiers, making them attractive for online reward-model updates (Cui et al., 2025a). Their appeal further comes from the fact that the log-ratio can in principle be decomposed into token-level increments and used to score the full next-token distribution at each step (Gao et al., 2025). However, using this decomposable sequence-level objective as local process supervision exposes a fundamental train–inference mismatch. During training, the log-ratio is constrained only through a sequence-level objective on terminal outcomes; at inference and during RL, it is decomposed and queried as token- or step-level scores for partial reasoning prefixes. As a result, local credits are only weakly identified: many different allocations of reward mass across tokens can satisfy the same final-outcome constraint, and the learned scores may capture formatting artifacts or other spurious correlations with success rather than faithfully reflecting local step quality.

---

[1]Sun Yat-sen University [2]Shenzhen Loop Area Institute [3]Meta AI [4]University of California, Davis. Correspondence to: Xiaojun Quan <xiaojunquan@slai.edu.cn>, Lifu Huang <lfuhuang@ucdavis.edu>.

*Proceedings of the 43rd International Conference on Machine Learning*, Seoul, South Korea. PMLR 306, 2026. Copyright 2026 by the author(s).

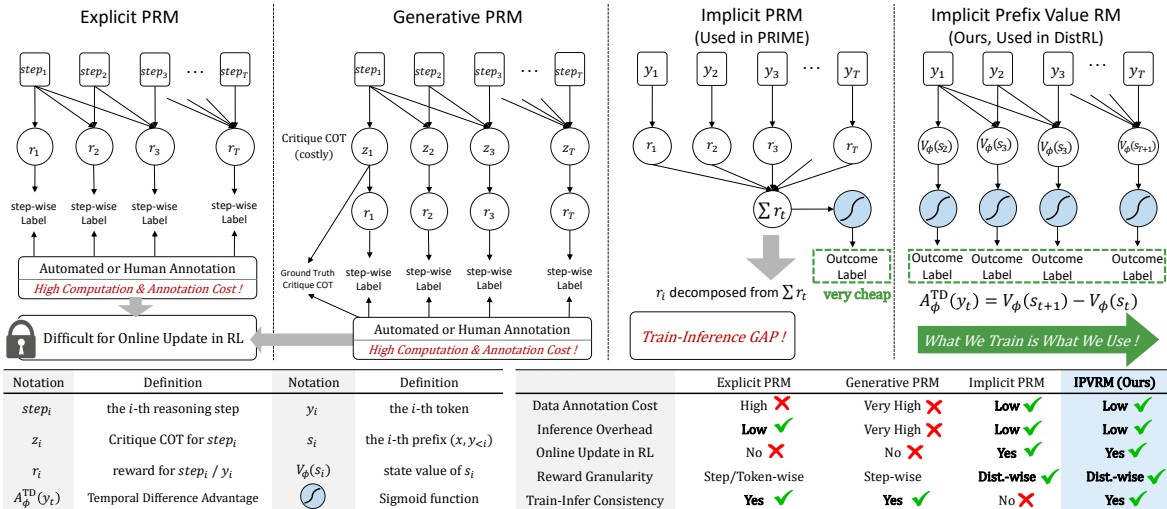

*Figure 1.* **Paradigm comparison and qualitative trade-offs.** Existing paradigms either incur high annotation/inference costs (Explicit/Generative PRMs) or suffer from a train–inference objective mismatch (Implicit PRMs). Our proposed IPVRM resolves these limitations by directly learning prefix-conditioned values $V_\phi(s_t)$, achieving both low-cost efficiency and training consistency ("what we train is what we use") to support Distribution-Level RL.

This concern is consistent with recent analyses of LM-induced implicit reward models, which show that log-likelihood-ratio rewards can generalize worse than explicit reward heads under distribution shifts and may rely heavily on superficial token-level cues rather than semantic quality (Lin et al., 2024; Razin et al., 2026). This fragility is directly relevant to process supervision in reasoning. In our experiments, implicit PRMs are trained on rollouts from a particular SFT policy, but are later expected to verify partial reasoning prefixes under different trace formats. As we examine on PROCESSBENCH in Section 4.2, this induces a format-shift scenario for step-level error localization: a model may still rank in-distribution SFT responses reasonably well, yet fail to verify reasoning steps when the trace format changes. The problem becomes more severe in distribution-level RL, where advantages are computed for many candidate tokens. If the underlying local scores are unreliable, dense candidate-token updates may amplify miscrediting by assigning positive advantages to superficially preferred but semantically incorrect continuations.

To address these challenges, we propose a unified framework that first makes implicit rewards faithful to the prefix-level quantities used during optimization, and then uses them to support distribution-level RL. We introduce the **Implicit Prefix-Value Reward Model (IPVRM)** to address the train–inference mismatch in prior implicit PRMs. Importantly, our critique is not that the log-likelihood-ratio parameterization is intrinsically unsuitable. Rather, IPVRM changes what the ratio is trained to represent: a prefix-conditioned value instead of an arbitrary token-level reward decomposition. Specifically, IPVRM directly learns a prefix-conditioned state value $V_\phi(s_t)$, which estimates the

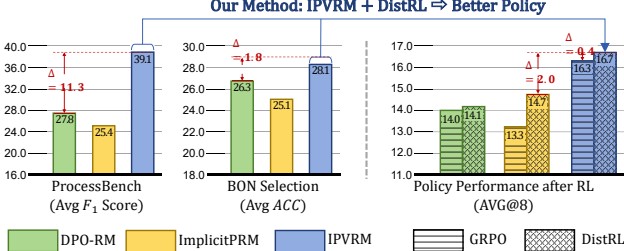

*Figure 2.* **Performance comparison on Qwen3-0.6B.** IPVRM improves both step-level error localization on PROCESSBENCH (Left) and sequence-level reranking in **Best-of-$N$ selection** (Middle) over prior implicit reward models. Leveraging these better signals, **DistRL** further outperforms standard **GRPO** in downstream policy performance (AVG@8) (Right).

probability of eventual correctness from the current partial reasoning prefix. This aligns the training target with the inference-time query: both require evaluating the quality of partial prefixes. Given these prefix values, we derive local optimization signals through temporal-difference advantages, $A_\phi^{\text{TD}} = V_\phi(s_{t+1}) - V_\phi(s_t)$. Since this TD signal is computed as a difference between prefix values, it measures how a continuation changes the predicted probability of eventual correctness, rather than directly treating token-level log-ratio scores as local correctness rewards. Moreover, this TD difference can be computed not only for the sampled token but also for high-probability candidate tokens, allowing the model to obtain candidate-level advantages without additional rollouts. Figure 1 illustrates how this design preserves the low annotation cost and online-update ability of implicit models while improving train–inference consistency.

Building on these prefix-level signals, we further introduce **Distribution-Level RL (DistRL)** as a lightweight extension to standard trajectory-level RL. DistRL extends the update from the sampled token to a truncated set of high-probability candidate tokens, using IPVRM-derived TD advantages to provide additional distribution-level supervision. In this way, the vocabulary-level scoring ability of implicit rewards is used not as an unreliable decomposition of a sequence-level objective, but as a prefix-value signal that complements trajectory-based policy optimization.

As summarized in Figure 2, IPVRM substantially improves process verification: on Qwen3-0.6B, it raises the average PROCESSBENCH $F_1$ from 27.8/25.4 with DPO-RM/Implicit PRM to 39.1, while also improving Best-of-$N$ reranking accuracy from 25.1/26.3 to 28.1. These better-aligned prefix-level signals translate into stronger downstream RL policies: GRPO with IPVRM improves AVG@8 from 14.0 with DPO-RM to 16.3, and DistRL further raises it to 16.7 by making limited but effective use of dense token-level scoring signals.

## 2. Preliminary

### 2.1. Notation and RL for Reasoning

We formulate autoregressive text generation with outcome-based supervision as a token-level Markov Decision Process (MDP). Given a prompt $x$, a policy $\pi_\theta$ generates a response $y = (y_1, \ldots, y_T)$ autoregressively. At step $t$, the state corresponds to the prefix $s_t = (x, y_{<t})$, the action is the token $y_t \in \mathcal{V}$, and the transition is deterministic with $s_{t+1} = s_t \oplus y_t$. Upon termination, a verifier assigns a binary outcome reward $r_o(x, y) \in \{0, 1\}$. Since supervision is outcome-only, the per-token reward is zero at all intermediate steps and only applied at termination: $r_o(s_t, y_t) = 0$ for $t < T$, and $r_o(s_T, y_T) = r_o(x, y)$. The RL objective is to maximize the expected return $J(\theta) = \mathbb{E}_{y \sim \pi_\theta}[r_o(x, y)]$. While this setup primarily targets reasoning tasks where verifiers check logical correctness, it generalizes to any generation task with terminal outcome labels. To estimate the quality of intermediate steps, we denote the state-value function as $V(s_t)$ and the state-action value function as $Q(s_t, y_t)$. The advantage function is typically defined as $A(s_t, y_t) = Q(s_t, y_t) - V(s_t)$, representing the relative improvement of action $y_t$ over the average expectation at $s_t$. During training, we collect trajectories using a behavior policy $\pi_{\text{old}}$ and regularize the student policy against a reference policy $\pi_{\text{ref}}$ (typically the SFT model). Throughout this paper, $\pi_\phi$ denotes the implicit reward model parameterized by $\phi$; $\rho_\theta(y_t) = \pi_\theta(y_t|s_t)/\pi_{\text{old}}(y_t|s_t)$ represents the importance sampling ratio; and $\sigma(\cdot)$ is the sigmoid function. Standard PPO hyperparameters $(\gamma, \lambda, \varepsilon)$ follow their conventional definitions.

### 2.2. Implicit Reward Models and Implicit PRM

Implicit Reward Models (IM-RMs) provide a convenient way to obtain reward signals from preference data. Unlike *explicit* reward models that predict rewards via an additional head over hidden representations, IM-RMs define rewards *implicitly* through the language model's own log probabilities. A canonical and widely used instantiation parameterizes the reward for a prompt–response pair $(x, y)$ as a log-likelihood ratio against a reference distribution $\pi_{\text{ref}}$ (typically the initialization of the trainable model):

$$r_\phi(x, y) = \beta \log \frac{\pi_\phi(y|x)}{\pi_{\text{ref}}(y|x)}, \tag{1}$$

where $\beta > 0$ is a temperature coefficient. Given a preference dataset $D = \{(x, y_w, y_\ell)\}$, where $y_w$ is preferred over $y_\ell$, a representative and widely adopted approach for learning such implicit rewards is Direct Preference Optimization (DPO) (Rafailov et al., 2023); meanwhile, other IM-RM training objectives have also been proposed (Yang et al., 2025b; Zhou et al., 2024; Meng et al., 2024). DPO optimizes a Bradley–Terry style objective that encourages a larger reward gap between preferred and dispreferred responses:

$$\mathcal{L}_{\text{DPO}}(\phi) = -\mathbb{E}_{(x, y_w, y_\ell) \sim D}\Big[\log \sigma\big(r_\phi(x, y_w) - r_\phi(x, y_\ell)\big)\Big]. \tag{2}$$

Once an IM-RM is trained and induces an implicit reward of the form in Equation (1), prior work (Zhong et al., 2025; Rafailov et al., 2024; Chen et al., 2026) further sharpens this sequence-level rewards into token-level rewards as:

$$r(s_t, y_t) = \beta \log \frac{\pi_\phi(y_t|s_t)}{\pi_{\text{ref}}(y_t|s_t)}, \tag{3}$$

which provides fine-grained reward for RL optimization.

Building on this, Implicit PRM (Yuan et al., 2025a) replaces pairwise preference supervision with outcome labels $r_o(x, y)$ and minimizes a binary cross-entropy objective:

$$\mathcal{L}_{\text{Implicit PRM}}(\phi) = -\mathbb{E}_{(x, y, r_o) \sim D}\Big[r_o(x, y) \cdot \log \sigma\Big(\sum_{t=1}^{T} r(s_t, y_t)\Big) \\ + (1 - r_o(x, y)) \cdot \log\Big[1 - \sigma\Big(\sum_{t=1}^{T} r(s_t, y_t)\Big)\Big]\Big]. \tag{4}$$

In this way, Implicit PRM can deliver process-level reward signals while requiring only sequence-level correctness labels. This reduces data collection costs compared to explicit PRMs that rely on step-level annotations (Wang et al., 2024) or pairwise datasets (Chen et al., 2025). The corresponding state-action value and next-state value induced by Implicit PRM can be written as

$$Q_\phi(s_t, y_t) = V_\phi(s_{t+1}) = \beta \sum_{i=1}^{t} \log \frac{\pi_\phi(y_i|s_i)}{\pi_{\text{ref}}(y_i|s_i)}, \tag{5}$$

which estimates the expected future return based on the cumulative prefix log-ratio up to step $t$. However, Implicit PRM may introduce

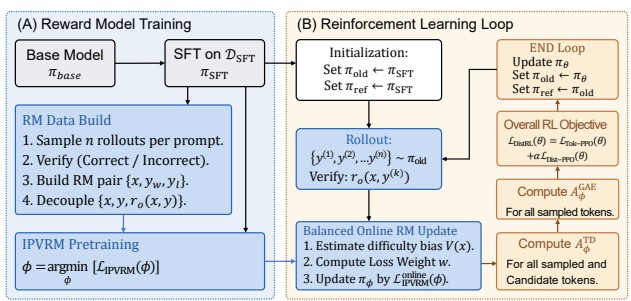

*Figure 3.* Overview of the IPVRM+DistRL framework.

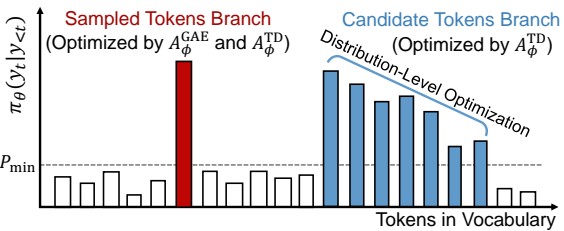

*Figure 4.* **Illustration of DistRL.** Unlike standard RL that optimizes solely on the sampled trajectory (red bar), DistRL introduces a **dual-branch** mechanism. It simultaneously updates high-probability candidate tokens (blue bars) using dense one-step TD advantages, while refining the sampled token (red bar) using a hybrid of GAE and TD advantages derived from IPVRM estimates.

a train–inference mismatch, as it regresses a sequence-level log-probability ratio through a sigmoid to fit the terminal correctness label during training, but applies prefix-level scoring at inference.

## 2.3. Advantage Estimation

In text-generation Markov Decision Processes (MDPs) with terminal rewards, advantage estimation propagates outcome supervision to earlier steps. Using the value function $V_\phi(s_t)$ (Eq. (5)), three key estimators are defined:

**Temporal-Difference (TD) Estimation.** The one-step TD advantage estimator is:

$$A_\phi^{\text{TD}}(s_t, y_t) = r_o(s_t, y_t) + \gamma V_\phi(s_{t+1}) - V_\phi(s_t). \quad (6)$$

It is computationally efficient but biased. The bias arises from reliance on a learned value function.

**Monte Carlo (MC) Estimation.** The MC advantage estimator uses full sequence-level return for token $y_t$:

$$A_\phi^{\text{MC}}(s_t, y_t) = r_o(x, y) - V_\phi(s_t). \quad (7)$$

It equals return minus the current prefix's baseline value, with low bias but high variance caused by delayed rewards.

**Generalized Advantage Estimation (GAE).** GAE balances TD/MC bias–variance trade-offs via interpolation (Schulman et al., 2015):

$$A_\phi^{\text{GAE}}(s_t, y_t) = \sum_{i=t}^{T} (\gamma\lambda)^{i-t} A_\phi^{\text{TD}}(s_i, y_i), \quad (8)$$

where $\lambda \in [0, 1]$ controls the trade-off. Aggregating token-level TD advantages with exponential weighting, GAE delivers stable estimation and robust optimization compared to TD/MC alone.

## 3. Methodology

We adopt a two-stage framework for RL-based reasoning shown in Figure 3. In Stage A (RM Training), we sample multiple rollouts per prompt from the SFT policy, verify outcomes, and train the **Implicit Prefix-Value Reward Model** (IPVRM) $\pi_\phi$ to estimate prefix values $V_\phi(s_t)$. The normalized sigmoid $\sigma(\bar{v}_\phi(t))$, where $\bar{v}_\phi(t) = V_\phi(s_t)/t$, is interpreted as the probability of eventual correctness from prefix $s_t$. In Stage B (DistRL Loop), we fine-tune the policy with **Distribution-Level RL** (DistRL): each iteration samples trajectories from $\pi_{\text{old}}$, computes token-level advantages $A_\phi^{\text{GAE}}$ on sampled tokens and dense distribution-level advantages $A_\phi^{\text{TD}}$ over vocabulary-wide candidate tokens, and updates $\pi_\theta$ with a hybrid PPO objective while updating IPVRM online.

### 3.1. Implicit Prefix-Value Reward Model

Standard Implicit PRMs suffer from a critical train–inference mismatch: the sequence-level objective constrains only the cumulative reward, so the model tends to "hide" reward mass on specific structural artifacts (e.g., fixed action tags) instead of distributing it according to reasoning. This creates a gap between what is optimized in training and what is needed at inference, making the model brittle when such patterns are absent during inference. As a result, it can still select correct sequences (high BoN accuracy) but fails to localize errors without these cues (low PROCESSBENCH $F_1$), leaving step-wise scores unreliable.

To resolve this, we propose IPVRM, which reframes the implicit reward as a prefix-conditioned state-value function. Rather than learning an underdetermined token-level decomposition whose cumulative sum explains the terminal outcome, IPVRM supervises each visited prefix to predict the likelihood that the current reasoning path will eventually reach a correct answer. Specifically, we define a prefix value $V_\phi(s_t)$ and apply the BCE objective to its length-normalized form $\bar{v}_\phi(t) = V_\phi(s_t)/t$, so that prefixes of different lengths are trained on a comparable scale. The normalized prefix value is regressed against the terminal outcome $r_o(x, y)$ with a margin $m$:

$$\mathcal{L}_{\text{IPVRM}}(\phi) = -\mathbb{E}_{(x,y,r_o)\sim D}\Big[\frac{1}{T}\sum_{t=1}^{T}\Big(r_o(x,y)\log\sigma\big(\bar{v}_\phi(t) - m\big)$$
$$+ \big(1 - r_o(x,y)\big)\log\Big[1 - \sigma\big(\bar{v}_\phi(t) + m\big)\Big]\Big)\Big]. \quad (9)$$

This objective turns outcome supervision into dense prefix-level training signals without introducing step annotations. Since every prefix along a trajectory is directly optimized toward the same eventual outcome, the learned value is encouraged to track the progress of partial reasoning paths rather than to concentrate on a few high-correlation tokens. At optimization time, each candidate token $y_t'$ is scored by how much it changes the learned prefix value, i.e., $V_\phi(s_t \oplus y_t') - V_\phi(s_t)$. Thus, the same quantity trained from outcome labels is directly used to judge whether a continuation improves the current reasoning path.

### 3.2. Distribution-Level RL (DistRL)

While standard RL algorithms (e.g., PPO or GRPO) are effective, they typically optimize only along the sampled token trajectory. This design does not fully exploit a key property of IM-RMs: they expose scores over the entire next-token distribution through the language-model head. DistRL leverages this property by adding

a **distribution-level update** over high-probability candidate tokens, while retaining the sampled trajectory as the main carrier of outcome-based credit assignment. As illustrated in Figure 4, this **Dual-Branch optimization** consists of the following components.

**Candidate Tokens Branch.** For candidate-token updates, we use the one-step change in IPVRM value as a dense local advantage. Based on Equation (6), with $\gamma = 1$ (Cui et al., 2025a; Rafailov et al., 2024) and $r_o(s_t, y_t) = 0$ for $t < T$ (outcome rewards are non-zero only at $\langle$EOS$\rangle$), the TD signal reduces to the value difference between two adjacent prefixes. For any candidate token $y'_t \in \mathcal{V}$, we compute

$$A_\phi^{\mathrm{TD}}(s_t, y'_t) = V_\phi(s_t \oplus y'_t) - V_\phi(s_t) = \beta \log \frac{\pi_\phi(y'_t|s_t)}{\pi_{\mathrm{old}}(y'_t|s_t)}. \quad (10)$$

Here we set $\pi_{\mathrm{ref}} = \pi_{\mathrm{old}}$ for the distribution-level update, so the advantage is measured relative to the behavior policy that generated the current batch. This keeps candidate-token optimization aligned with the on-policy distribution and reduces variance compared with a fixed SFT reference (see Section B). In practice, we standardize $V_\phi$ within each minibatch before computing TD differences (Section C), and restrict the update to a truncated candidate set $\mathcal{C} = \{ y'_t \in \mathcal{V} : \pi_{\mathrm{old}}(y'_t|s_t) \geq P_{\min} \}$, which filters out tail noise while retaining most of the policy mass. Finally, let $\rho_\theta(y'_t) = \frac{\pi_\theta(y'_t|s_t)}{\pi_{\mathrm{old}}(y'_t|s_t)}$ denote the importance sampling ratio, the *Distribution-level PPO* objective is then defined as the clipped expectation of TD advantages over $\mathcal{C}$:

$$\mathcal{L}_{\mathrm{Dist\text{-}PPO}}(\theta) = -\mathbb{E}_{y \sim \pi_{\mathrm{old}}} \Big[ \frac{1}{T} \sum_{t=1}^{T} \sum_{y'_t \in \mathcal{C}} \pi_{\mathrm{old}}(y'_t|s_t) \min \Big($$
$$\rho_\theta(y'_t) A_\phi^{\mathrm{TD}}(s_t, y'_t), \mathrm{CLIP}(\rho_\theta(y'_t), \varepsilon) A_\phi^{\mathrm{TD}}(s_t, y'_t) \Big) \Big]. \quad (11)$$

Sensitivity to $m$ and $P_{\min}$ is reported in Appendix D.

**Sampled Tokens Branch.** Although TD advantages provide dense reward signals, they do not capture long-horizon returns and therefore cannot fully replace trajectory-based estimators. Consequently, for the sampled token $y_t$, we retain a trajectory-based GAE estimator together with a reliable rule-based outcome reward $r_o$, and optimize the conventional GRPO-style surrogate:

$$\mathcal{L}_{\mathrm{Tok\text{-}PPO}}(\theta) = -\mathbb{E}_{y \sim \pi_{\mathrm{old}}} \Big[ \frac{1}{T} \sum_{t=1}^{T} \min \Big( \rho_\theta(y_t) A(s_t, y_t),$$
$$\mathrm{CLIP}(\rho_\theta(y_t), \varepsilon) A(s_t, y_t) \Big) \Big]. \quad (12)$$

Here, the overall advantage $A(s_t, y_t)$ combines a group-normalized outcome signal (as in GRPO) with the token-level GAE advantage estimated by the IM-RM. Concretely, for each prompt $x$, we draw $n$ rollouts $\{y^{(k)}\}_{k=1}^{n} \sim \pi_{\mathrm{old}}(\cdot|x)$ and define the per-prompt mean and (empirical) standard deviation of outcome rewards:

$$\mu(x) = \frac{1}{n} \sum_{k=1}^{n} r_o(x, y^{(k)}), \; s(x) = \sqrt{\frac{1}{n} \sum_{k=1}^{n} \big(r_o(x, y^{(k)}) - \mu(x)\big)^2}.$$

In implementation, we compute $A_\phi^{\mathrm{GAE}}$ by first standardizing TD advantages across the $n$ rollouts for the same prompt (see Appendix C) and then applying Equation (8). Then, for a sampled trajectory $y = y^{(k)}$ and its token $y_t$ at state $s_t = (x, y_{<t})$, we set

$$A(s_t, y_t) = \frac{r_o(x, y) - \mu(x)}{s(x)} + A_\phi^{\mathrm{GAE}}(s_t, y_t). \quad (13)$$

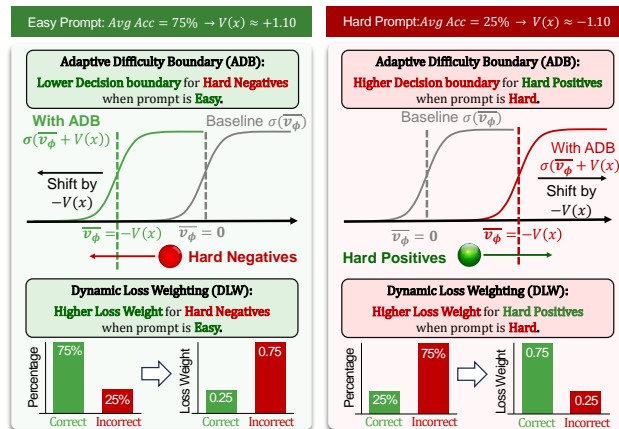

*Figure 5.* Schematic of Adaptive Difficulty Boundary (ADB) and Dynamic Loss Weighting (DLW). As illustrated, ADB shifts the sigmoid decision boundary by $V(x)$ (the expected accuracy), while DLW reweights samples. This focuses learning on *hard negatives* for easy prompts (left) and *hard positives* for hard prompts (right), mitigating label imbalance.

**Overall DistRL Objective.** The final DistRL objective combines these two branches by a hyperparameter $\alpha$ as:

$$\mathcal{L}_{\mathrm{DistRL}}(\theta) = \mathcal{L}_{\mathrm{Tok\text{-}PPO}}(\theta) + \alpha \, \mathcal{L}_{\mathrm{Dist\text{-}PPO}}(\theta) \quad (14)$$

This enables a single rollout to provide dense learning signals for multiple plausible next tokens, effectively complementing trajectory-only updates.

### 3.3. Online RM Update with Adaptive Balancing

To prevent the reward model from becoming stale as the policy shifts, we update IPVRM online using the rollouts from $\pi_{\mathrm{old}}$. However, online data often suffers from extreme label imbalance (e.g., easy prompts yield mostly correct answers). We address this via two mechanisms visualized in Figure 5. First, **Adaptive Difficulty Boundary (ADB)** incorporates the prompt's initial state value $V(x) = \mathrm{logit}(\mu(x))$ as a dynamic baseline, shifting the sigmoid boundary to focus on "hard" examples (e.g., incorrect answers on easy prompts). Second, **Dynamic Loss Weighting (DLW)** rebalances the gradient contribution $w$ based on the rarity of the outcome (specifically, setting $w = 1 - \mu(x)$ for correct responses and $w = \mu(x)$ for incorrect ones). Overall, the balanced online objective is:

$$\mathcal{L}_{\mathrm{IPVRM}}^{\mathrm{online}}(\phi) = -\mathbb{E}_{(x,y,r_o) \sim \pi_{\mathrm{old}}} \Big[ \frac{w}{T} \sum_{t=1}^{T} \Big($$
$$r_o(x, y) \cdot \log \sigma \big(\bar{v}_\phi(t) - m + V(x)\big)$$
$$+ (1 - r_o(x, y)) \cdot \log \Big[1 - \sigma \big(\bar{v}_\phi(t) + m + V(x)\big)\Big] \Big) \Big]. \quad (15)$$

This design stabilizes online IPVRM updates under label imbalance and improves sample efficiency by focusing learning on informative (hard) outcomes within each prompt.

## 4. Experiments and Analysis

We evaluate our method on complex reasoning tasks with verifiable final answers, which allow objective measurement of both reward quality and downstream RL performance. We first assess whether

**IPVRM** yields more reliable reward signals than prior reward models, and then test whether these improvements lead to stronger policy optimization with **DistRL**.

## 4.1. Experimental Setup

**Models.** We use **Qwen3-0.6B/8B-Base** (Yang et al., 2025a) and **Llama-3.2-1B/3B** (Dubey et al., 2024) as backbone models. These four backbones cover both Qwen and Llama families across small-to-medium scales, enabling a systematic study of reward modeling and RL behavior under a feasible compute budget.

**Datasets.** We use the **PRIME-RL/Eurus-2** SFT and RL corpora. For initialization, we use **Eurus-2-SFT-Data**[1] ($\sim$230K), an action-tagged chain-of-thought dataset spanning math, code, and science. It structures reasoning into explicit steps such as ASSESS, ADVANCE, and VERIFY, which is convenient for studying process supervision. For RL, we use **Eurus-2-RL-Data**[2] ($\sim$480K train), which contains competition-style math and coding problems equipped with deterministic verifiers, such as boxed LATEX answers for mathematics.

**Pipeline.** We follow a three-stage pipeline. First, we perform supervised fine-tuning (SFT) to obtain the initial policy $\pi_{\text{SFT}}$. We then construct RM training data by sampling rollouts from $\pi_{\text{SFT}}$: for each prompt, we generate five trajectories (temperature 1.0), verify them against ground truth, retain prompts with mixed outcomes, and form one (correct, incorrect) pair. This ensures that the RM is trained on the policy's own distribution. Finally, we train and evaluate the RM, and use it to provide dense rewards for RL.

**Benchmarks.** We test both reward models and policy models.

For *RM evaluation*, we use two complementary settings. First, we evaluate sequence-level ranking with **Best-of-$N$** reranking, where $N \in \{4, 16, 64\}$, on **MATH-500** (Hendrycks et al., 2021) and **Minerva-Math** (Lewkowycz et al., 2022), with AMC-23 results reported in Section E. In this setting, all candidate responses are sampled from the SFT policy. Second, we evaluate process-level verification on **PROCESSBENCH** (Zheng et al., 2025), reported by $F_1$, which measures whether an RM can localize erroneous reasoning steps.

These two settings differ in their relation to the RM training distribution. Since the RM is trained on SFT-policy rollouts for subsequent RL, Best-of-$N$ reranking over candidates sampled from the same SFT policy serves as an in-distribution evaluation of sequence-level ranking. In contrast, PROCESSBENCH uses fixed-format reasoning traces that differ from the structured SFT rollouts used for RM training, and therefore serves as a format-shift out-of-distribution evaluation of step-level error localization.

For *policy evaluation*, we report the final **avg@8** accuracy on **AIME** (2022–2024), **OlympiadBench** (He et al., 2024), **AMC**, **MATH-500**, and **Minerva-Math**. Avg@8 is computed by sampling eight responses per problem with temperature 0.6 and top-$p$ 0.95, and averaging their correctness. For OlympiadBench, we evaluate only the OE_TO_maths_en_COMP subset, consisting of 675 English plain-text open-ended mathematical competition problems. These benchmarks assess whether better reward signals lead

---

[1] https://huggingface.co/datasets/PRIME-RL/Eurus-2-SFT-Data
[2] https://huggingface.co/datasets/PRIME-RL/Eurus-2-RL-Data

*Table 1.* Different reward models' best-of-N sampling performance on MATH-500 and Minerva-Math with SFT models (Avg is computed by averaging over the two datasets, where each dataset score is the mean over @4/@16/@64).

| Base Model | RM Method | MATH-500 | | | Minerva-Math | | | Avg. |
|---|---|---|---|---|---|---|---|---|
| | | @4 | @16 | @64 | @4 | @16 | @64 | |
| Qwen3-0.6B Base | Q-RM | 41.4 | 44.4 | 44.6 | 9.2 | **12.1** | 10.7 | 27.1 |
| | EndoRM | 32.0 | 33.8 | 38.8 | 6.3 | 8.8 | 7.7 | 21.2 |
| | DPO-RM | 41.6 | 42.6 | 42.2 | **10.7** | 9.9 | 11.0 | 26.3 |
| | Implicit PRM | 39.6 | 39.6 | 38.6 | 10.3 | 11.4 | 11.0 | 25.1 |
| | IPVRM | **42.0** | **46.0** | **47.6** | 10.3 | 11.0 | **11.4** | **28.1** |
| Qwen3-8B Base | Q-RM | **66.8** | **69.0** | 68.8 | **26.5** | **27.0** | 25.7 | **47.3** |
| | EndoRM | 60.6 | 61.0 | 57.6 | 19.5 | 22.8 | 21.7 | 40.5 |
| | DPO-RM | 66.6 | 68.0 | 69.8 | 24.3 | 24.6 | **26.8** | 46.7 |
| | Implicit PRM | 66.4 | 68.0 | **70.2** | 24.6 | 24.3 | 26.5 | 46.7 |
| | IPVRM | **66.8** | 67.6 | 69.2 | 25.4 | 25.0 | **26.8** | 46.8 |
| Llama3.2-1B | Q-RM | 13.6 | **14.6** | 13.4 | 2.2 | 2.6 | 3.3 | 8.3 |
| | EndoRM | 10.4 | 10.6 | 10.0 | 1.1 | 1.8 | 2.2 | 6.0 |
| | DPO-RM | 11.2 | 12.6 | 12.6 | **3.3** | **3.7** | 2.9 | 7.7 |
| | Implicit PRM | 13.0 | 14.0 | 14.2 | 2.9 | 2.6 | 3.7 | 8.4 |
| | IPVRM | **13.9** | 14.2 | **14.8** | 1.5 | 2.6 | **4.4** | **8.6** |
| Llama3.2-3B | Q-RM | 29.0 | 31.4 | 33.8 | 5.9 | **9.9** | 8.1 | 19.7 |
| | EndoRM | 25.4 | 26.8 | 27.0 | 5.9 | 4.8 | 5.5 | 15.9 |
| | DPO-RM | 29.2 | 31.6 | 34.6 | 6.3 | **9.9** | 7.7 | 19.9 |
| | Implicit PRM | 29.8 | 32.8 | 35.4 | 5.9 | 9.2 | **8.5** | 20.3 |
| | IPVRM | **30.6** | **34.6** | **35.8** | **7.0** | 8.5 | 7.7 | **20.7** |

to stronger downstream mathematical reasoning after RL.

## 4.2. Evaluating the Reward Models

We first assess the quality of IPVRM as a reward model, covering both sequence-level ranking and process-level error localization. We then conduct ablations to analyze how its prefix-value objective and training dynamics contribute to reward reliability.

### 4.2.1. MAIN RESULTS OF RMS

**Baselines.** We compare IPVRM against a diverse set of established reward models spanning both explicit and implicit paradigms, including **Q-RM** (Chen et al., 2025), **EndoRM** (Li et al., 2025), **DPO-RM** (Rafailov et al., 2023), and **Implicit PRM** (Yuan et al., 2025a). These baselines are evaluated under both sequence-level reranking (Best-of-$N$) and fine-grained step verification (PROCESSBENCH).

For PROCESSBENCH, prior work (Yuan et al., 2025a)[3] interprets the sigmoid of the prefix log-likelihood ratio, $\sigma\big(\beta \sum_{i=1}^{t_1} \log \frac{\pi_\phi(y_i|s_i)}{\pi_{\text{ref}}(y_i|s_i)}\big)$, as the probability that a reasoning step is correct, where $t_1$ denotes the index of the last token of the current process. In the main results, we instead use a process-level log-likelihood ratio, $\sigma\big(\beta \sum_{i=t_2}^{t_1} \log \frac{\pi_\phi(y_i|s_i)}{\pi_{\text{ref}}(y_i|s_i)}\big)$, where $t_2$ is the index of the first token of the current process. This formulation isolates the current reasoning step rather than accumulating evidence from earlier prefixes, and yields consistently higher $F_1$ across all IM-RMs. Appendix G reports both variants.

**Analysis.** Table 1 and Table 2 together support the central motivation of IPVRM. On **Best-of-$N$** reranking, IPVRM generally matches or improves over prior implicit reward models across datasets and backbones, while remaining competitive with the strong explicit baseline Q-RM. The gains are most visible on

---

[3] https://github.com/PRIME-RL/ImplicitPRM/tree/main/eval

*Table 2.* Performance ($F_1$ score) of different reward modeling methods on PROCESSBENCH.

| Base Model | RM Method | GSM8K | MATH | Olympiad Bench | Omni Math | Avg. |
|---|---|---|---|---|---|---|
| **Qwen3-0.6B base** | Q-RM | 47.2 | 38.9 | **31.7** | **35.0** | 38.2 |
| | EndoRM | 33.7 | 30.1 | 26.9 | 31.5 | 30.6 |
| | DPO-RM | 32.1 | 32.7 | 23.5 | 22.9 | 27.8 |
| | Implicit PRM | 30.3 | 31.0 | 21.5 | 18.8 | 25.4 |
| | IPVRM | **52.8** | **42.1** | 30.0 | 31.3 | **39.1** |
| **Qwen3-8B base** | Q-RM | **61.0** | 45.2 | **39.5** | 35.3 | 45.3 |
| | EndoRM | 35.0 | 30.9 | 28.1 | 29.3 | 30.8 |
| | DPO-RM | 52.2 | **48.2** | 36.0 | 33.3 | 42.4 |
| | Implicit PRM | 51.6 | 47.8 | 36.4 | 33.1 | 42.2 |
| | IPVRM | 57.2 | 47.8 | 38.8 | **38.6** | **45.6** |
| **Llama3.2-1B** | Q-RM | 29.7 | 30.4 | 22.9 | 26.3 | 27.3 |
| | EndoRM | 33.5 | 29.2 | 26.0 | 32.0 | 30.2 |
| | DPO-RM | 38.7 | 27.9 | 21.2 | 22.0 | 27.5 |
| | Implicit PRM | 37.8 | 28.9 | 21.4 | 23.5 | 27.9 |
| | IPVRM | **42.7** | **35.2** | **28.7** | **29.2** | **34.0** |
| **Llama3.2-3B** | Q-RM | 42.7 | 37.5 | **34.4** | **36.1** | 37.7 |
| | EndoRM | 32.5 | 28.9 | 23.9 | 29.7 | 28.8 |
| | DPO-RM | 41.2 | 33.9 | 25.1 | 27.4 | 31.9 |
| | Implicit PRM | 39.9 | 33.9 | 23.4 | 22.9 | 30.0 |
| | IPVRM | **48.1** | **38.7** | 32.1 | 34.2 | **38.3** |

*Table 3.* Variants of IPVRM: best-of-N sampling performance on MATH-500 with SFT model.

| Base Model | RM Method | MATH-500 | | | Avg. |
|---|---|---|---|---|---|
| | | @4 | @16 | @64 | |
| **Qwen3-0.6B base** | IPVRM$_{\text{late}}$ | **42.4** | **47.0** | **50.0** | **46.5** |
| | IPVRM | 42.0 | 46.0 | 47.6 | 45.2 |
| | IPVRM$_{\text{early}}$ | 41.0 | 44.2 | 44.2 | 43.1 |
| **Llama3.2-1B base** | IPVRM$_{\text{late}}$ | **14.4** | **14.6** | **17.0** | **15.3** |
| | IPVRM | 13.9 | 14.0 | 16.4 | 14.8 |
| | IPVRM$_{\text{early}}$ | 13.6 | 14.2 | 14.8 | 14.2 |

*Table 4.* Performance ($F_1$ score) of different IPVRM variants on PROCESSBENCH.

| Base Model | RM Method | GSM8K | MATH | Olympiad Bench | Omni Math | Avg. |
|---|---|---|---|---|---|---|
| **Qwen3-0.6B base** | IPVRM$_{\text{late}}$ | 50.8 | 40.3 | 28.4 | 29.2 | 37.2 |
| | IPVRM | **52.6** | 41.9 | 30.0 | 31.5 | 39.0 |
| | IPVRM$_{\text{early}}$ | 49.9 | **43.4** | **32.2** | **34.8** | **40.1** |
| **Llama3.2-1B base** | IPVRM$_{\text{late}}$ | 41.1 | 33.8 | 27.1 | 27.6 | 32.4 |
| | IPVRM | 42.7 | 35.2 | 28.7 | 29.2 | 34.0 |
| | IPVRM$_{\text{early}}$ | **45.0** | 34.8 | **30.1** | **31.4** | **35.3** |

smaller backbones, where reward modeling is harder. For example, on Qwen3-0.6B, IPVRM improves the average BoN score from 25.1 to 28.1 over Implicit PRM. On larger backbones such as Qwen3-8B, it remains highly competitive (e.g., 46.8 vs. 46.7 average BoN over Implicit PRM), showing that IPVRM preserves strong sequence-level ranking on in-distribution candidate responses.

**PROCESSBENCH** provides the more revealing test. Unlike BoN, it evaluates whether the model can localize reasoning errors in traces whose format differs substantially from the structured SFT trajectories used for RM training. In this setting, prior implicit sequence-level objectives such as DPO-RM and Implicit PRM underperform both Q-RM and IPVRM, despite being competitive in BoN. This gap suggests that sequence-level supervision alone can overfit to structural regularities in SFT rollouts and may not transfer reliably to step-level verification under format shift. By directly supervising prefix-conditioned values, IPVRM enforces prefix–outcome consistency and yields substantially stronger error localization across backbones (e.g., +11.3 $F_1$ over DPO-RM on Qwen3-0.6B). Overall, these results support our claim that IPVRM better aligns reward-model training with inference-time usage.

### 4.2.2. ABLATION ON TEMPORAL WEIGHTING OF PREFIX LOSSES

IPVRM supervises every prefix with a value-style objective. This raises the question of which parts matter most. We study this by reweighting losses to emphasize early or late prefixes. We first rewrite the objective in Equation (9) at a single timestep $t$ as:

$$
\mathcal{L}_{\text{IPVRM-single}}(\phi, s_t, y_t) = -\Big[ r_o(x, y) \cdot \log \sigma(\bar{v}_\phi(t) - m) \\ + (1 - r_o(x, y)) \cdot \log(1 - \sigma(\bar{v}_\phi(t) + m)) \Big]. \quad (16)
$$

Based on this single per-prefix loss, we define two variants. **IPVRM$_{\text{late}}$** assigns larger weights to later timesteps, while **IPVRM$_{\text{early}}$** assigns larger weights to earlier timesteps. The standard IPVRM uses uniform weights.

$$
\mathcal{L}_{\text{IPVRM}_{\text{late}}}(\phi) = \frac{\sum_{t=1}^{T} \frac{t}{T} \mathcal{L}_{\text{IPVRM-single}}(\phi, s_t, y_t)}{\sum_{t=1}^{T} \frac{t}{T}}, \quad (17)
$$

$$
\mathcal{L}_{\text{IPVRM}_{\text{early}}}(\phi) = \frac{\sum_{t=1}^{T} \left(1 - \frac{t}{T}\right) \mathcal{L}_{\text{IPVRM-single}}(\phi, s_t, y_t)}{\sum_{t=1}^{T} \left(1 - \frac{t}{T}\right)}. \quad (18)
$$

We evaluate these variants on both sequence-level reranking and process-level verification (results in Tables 3 and 4).

The results suggest that different prefix regions matter for different downstream objectives. On **BoN**, IPVRM$_{\text{late}}$ performs best, indicating that upweighting later prefixes is more helpful for whole-trajectory reranking, which is consistent with BoN's goal of selecting a completed response. On **PROCESSBENCH**, IPVRM$_{\text{early}}$ achieves the best $F_1$, suggesting that emphasizing earlier prefixes better supports process-level discrimination, where off-track reasoning should be identified as early as possible. Overall, these trends are consistent with the prefix-value interpretation of IPVRM: later prefixes contribute more to final-response ranking, whereas earlier prefixes matter more for early error localization. The standard uniform objective remains the most balanced default across both settings.

### 4.3. Evaluating Distribution-Level RL (DistRL)

After establishing that IPVRM provides stronger and more robust reward signals, we evaluate how these improvements affect downstream RL. We also assess the effectiveness of DistRL and analyze the contributions of different components through ablations.

#### 4.3.1. MAIN RESULTS OF POLICY MODELS

For policy optimization, we compare our proposed **DistRL** against representative outcome- or process-based RL baselines, including **GRPO** (Shao et al., 2024), **PRIME** (Cui et al., 2025a), **SPRO** (Fei et al., 2025), and **Reinforce w/ Q-RM** (Chen et al., 2025). For the *GRPO w/ PRMs* setting, we use Equation (12) as the training objective, combining group-normalized outcome rewards with PRM-derived GAE advantages.

Results in Table 5 show that our method is competitive with, and often outperforms, representative RL baselines across model scales. Compared with outcome- or process-based methods such as **GRPO**, **PRIME**, **SPRO**, and **Reinforce w/ Q-RM**, **DistRL w/**

*Table 5.* Main results for policy model (AVG@8)

| Base Model | Method | AIME | MATH 500 | Minerva Math | Olympiad Bench | AMC | Avg. |
|---|---|---|---|---|---|---|---|
| Qwen3-0.6B base | SFT | 0.4 | 33.2 | 7.3 | 7.6 | 9.3 | 11.6 |
| | PRIME | 1.1 | 39.6 | 11.4 | 11.4 | 12.3 | 15.2 |
| | SPRO | **1.7** | 38.6 | 10.7 | 11.6 | 13.7 | 15.3 |
| | Reinforce w/ Q-RM | 1.1 | 36.4 | 9.2 | 11.0 | 10.8 | 13.7 |
| | GRPO w/ VR | 0.8 | 38.3 | 10.5 | 10.5 | 12.5 | 14.5 |
| | GRPO w/ DPO-RM | 0.8 | 38.4 | 9.8 | 10.0 | 11.1 | 14.0 |
| | GRPO w/ IPVRM | 1.2 | 41.5 | 12.1 | **12.9** | 13.9 | 16.3 |
| | DistRL w/ IPVRM | 1.5 | **42.5** | **13.3** | 12.1 | **14.0** | **16.7** |
| Qwen3-8B base | SFT | 3.1 | 60.8 | 30.9 | 20.6 | 28.3 | 28.7 |
| | PRIME | 5.7 | 67.9 | 34.3 | 30.8 | 33.4 | 34.4 |
| | SPRO | 5.6 | 68.7 | 35.8 | 32.2 | 34.0 | 35.3 |
| | Reinforce w/ Q-RM | 3.6 | 65.5 | 34.2 | 32.2 | 31.6 | 33.4 |
| | GRPO w/ VR | 5.7 | 65.4 | 33.5 | 29.7 | 33.4 | 33.5 |
| | GRPO w/ DPO-RM | 4.2 | 67.9 | 36.4 | 31.8 | 31.3 | 34.3 |
| | GRPO w/ IPVRM | 6.7 | **70.9** | 35.2 | 33.8 | 35.1 | 36.3 |
| | DistRL w/ IPVRM | **8.2** | 70.7 | **38.4** | **34.1** | **37.3** | **37.7** |
| Llama3.2-1B | SFT | 0.3 | 10.7 | 1.7 | 2.2 | 2.9 | 3.6 |
| | PRIME | 0.1 | 11.5 | 2.4 | 2.0 | 3.6 | 3.9 |
| | SPRO | 0.1 | 11.4 | 3.2 | 2.5 | **5.0** | 4.4 |
| | Reinforce w/ Q-RM | 0.1 | 10.8 | 2.5 | 2.5 | **5.0** | 4.2 |
| | GRPO w/ VR | 0.0 | 11.0 | 3.2 | 2.5 | 4.2 | 4.2 |
| | GRPO w/ DPO-RM | **0.4** | 11.2 | 3.2 | 2.8 | 4.8 | 4.5 |
| | GRPO w/ IPVRM | 0.1 | **13.2** | 2.6 | 3.0 | 4.5 | 4.7 |
| | DistRL w/ IPVRM | 0.1 | **13.2** | 3.4 | 3.1 | 4.2 | **4.8** |
| Llama3.2-3B | SFT | 0.4 | 27.5 | 6.0 | 5.6 | 7.1 | 9.3 |
| | PRIME | 0.4 | 26.6 | 7.5 | 5.2 | 6.5 | 9.2 |
| | SPRO | 0.1 | 27.2 | 5.6 | 5.4 | 4.5 | 8.6 |
| | Reinforce w/ Q-RM | 0.1 | 27.4 | 5.9 | 5.1 | 7.7 | 9.2 |
| | GRPO w/ VR | 0.1 | **29.8** | 6.8 | 5.6 | 6.9 | 9.8 |
| | GRPO w/ DPO-RM | 0.3 | 26.5 | 5.6 | 5.1 | 6.6 | 8.8 |
| | GRPO w/ IPVRM | 0.4 | 26.9 | **8.6** | 5.7 | 8.1 | 9.9 |
| | DistRL w/ IPVRM | **0.6** | 26.6 | 8.4 | 6.5 | 9.2 | **10.3** |

*Table 6.* Impact of reward models and DistRL across mathematical benchmarks (AVG@8).

| Base Model | Method | AIME | MATH 500 | Minerva Math | Olympiad Bench | AMC | Avg. |
|---|---|---|---|---|---|---|---|
| Qwen3-0.6B base | GRPO w/ DPO-RM | 0.8 | 38.4 | 9.8 | 10.0 | 11.1 | 14.0 |
| | **DistRL** w/ DPO-RM | 0.8 | 37.2 | 9.4 | 10.1 | 13.0 | 14.1 |
| | GRPO w/ Implicit PRM | 1.2 | 34.9 | 10.2 | 9.5 | 10.8 | 13.3 |
| | **DistRL** w/ Implicit PRM | 0.8 | 37.1 | 10.9 | 10.5 | **14.2** | 14.7 |
| | GRPO w/ IPVRM | 1.2 | 41.5 | 12.1 | **12.9** | 13.9 | 16.3 |
| | **DistRL** w/ **IPVRM** | 1.5 | **42.5** | 13.3 | 12.1 | 14.0 | **16.7** |
| Llama3.2-1B base | GRPO w/ DPO-RM | **0.4** | 11.2 | 3.2 | 2.8 | 4.8 | 4.5 |
| | **DistRL** w/ DPO-RM | 0.3 | 12.5 | 2.8 | 2.7 | **5.0** | 4.7 |
| | GRPO w/ Implicit PRM | 0.3 | 12.8 | 2.8 | 2.5 | 3.9 | 4.5 |
| | **DistRL** w/ Implicit PRM | **0.4** | **13.7** | 3.1 | 2.4 | 3.9 | 4.7 |
| | GRPO w/ IPVRM | 0.1 | 13.2 | 2.6 | 3.0 | 4.5 | 4.7 |
| | **DistRL** w/ **IPVRM** | 0.1 | 13.2 | **3.4** | **3.1** | 4.2 | **4.8** |

**IPVRM** achieves the best average performance on all four backbones. For example, on Qwen3-0.6B, it improves the average score from 15.2–15.3 for PRIME/SPRO to 16.7, and on Qwen3-8B, it reaches 37.7, outperforming all compared baselines. These results show that the proposed framework delivers strong overall policy improvement across both Qwen and Llama backbones.

### 4.3.2. ABLATION ON DIFFERENT IM-RMS WITH DISTRL

To disentangle the impact of reward quality and RL optimization, we compare different combinations of implicit reward models and policy optimizers, with a focus on IPVRM and DistRL against standard baselines.

The results in Table 6 show two consistent trends. First, IPVRM is the strongest reward model for downstream RL: under both GRPO and DistRL, it generally yields the highest final reasoning accuracy. For example, on Qwen3-0.6B, DistRL w/ IPVRM reaches 16.7, outperforming DistRL w/ DPO-RM (14.1) and DistRL w/ Implicit PRM (14.7). This indicates that better prefix-value modeling translates into better downstream policy optimization. Second, DistRL consistently improves over trajectory-only updates with the same reward model, typically by 0.1–1.4 points. This suggests that DistRL mainly serves to better exploit a fixed reward

*Table 7.* Training-time analysis of TD. DistRL achieves higher RM score and verifier accuracy than GRPO across all backbones. Each entry is reported as **DistRL / GRPO**.

| | RM score | Verifier acc. |
|---|---|---|
| Qwen3-0.6B | $-0.140$ / $-0.181$ | 23.5 / 23.0 |
| Qwen3-8B | $-0.178$ / $-0.202$ | 41.6 / 41.4 |
| Llama3.2-1B | $-0.040$ / $-0.050$ | 7.7 / 7.4 |
| Llama3.2-3B | $-0.024$ / $-0.025$ | 15.9 / 15.5 |

model by leveraging unsampled candidate tokens. The strongest combination remains IPVRM + DistRL.

### 4.3.3. ANALYSIS OF TD ADVANTAGE

A key question for DistRL is whether its one-step TD signal reflects long-horizon return. To study this, we conduct an analysis on **MATH-500** using the Qwen3-0.6B SFT policy with IPVRM. For each sampled trajectory, we randomly select a truncation point, enumerate the top-5 candidate next tokens, and compare the one-step TD signal $A_\phi^{\mathrm{TD}}(s_t, y_t')$ with the Monte Carlo outcome obtained by rolling out from $s_t \oplus y_t'$ to completion.

Across 2,500 candidate branches, the accumulated prefix value retains moderate predictive power for final correctness: the absolute prefix value $|V_\phi(s_t \oplus y_t')|$ achieves an AUC-ROC of 0.64. In contrast, the correlation between one-step TD and single-rollout binary outcomes is very weak (Pearson 0.0226), which is expected in long-horizon reasoning where final success is rarely determined by a single token. This suggests that $A_\phi^{\mathrm{TD}}$ should not be interpreted as a Monte Carlo estimator of long-horizon return.

Instead, TD in DistRL serves as a *local optimization signal* that biases the policy toward candidate continuations preferred by the reward model. To examine this mechanism, we analyze two training-time metrics: the mean per-step RM score,

$$\mathrm{RM\ Score}(x, y) = \frac{1}{|y|} \left( \log \pi_\phi(y \mid x) - \log \pi_{\mathrm{ref}}(y \mid x) \right), \quad (19)$$

and the rollout verifier accuracy, both averaged over training trajectories.

As shown in Table 7, DistRL consistently achieves higher mean per-step RM scores and higher verifier accuracy than GRPO across all backbones. This indicates that TD-based updates guide the policy toward trajectories that are more strongly preferred by the reward model and more likely to be correct.

Overall, these results clarify the role of TD in DistRL: although it is not a faithful estimator of long-horizon return, it provides a useful local signal that improves alignment with the reward model. This offers a plausible explanation for the consistent policy gains of DistRL over GRPO observed in Table 6.

### 4.3.4. ABLATION ON RM UPDATE WITH ADAPTIVE BALANCING

Since IPVRM is updated online during RL, a key practical question is whether the RM remains stable as the policy distribution shifts. We therefore evaluate the contribution of our balancing strategy for online RM updates.

We compare five configurations on Llama3.2-1B: **Frozen RM**,

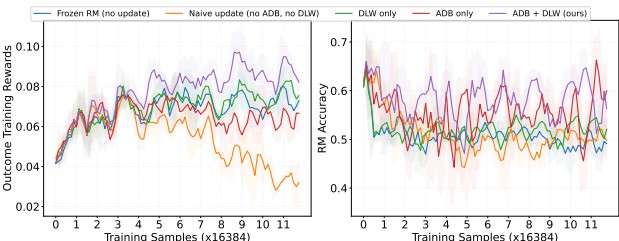

*Figure 6.* Ablation of online IPVRM update strategies on Llama3.2-1B. Left: outcome training reward during RL. Right: RM accuracy across training samples ($\times 16384$).

*Table 8.* Policy performance under different online RM update settings on Qwen3-0.6B with DistRL.

| Method | AIME | MATH500 | Minerva | Olympiad | AMC | Avg |
|---|---|---|---|---|---|---|
| Implicit PRM w/o ADB/DLW | 0.8 | 37.1 | 10.9 | 10.5 | 14.2 | 14.7 |
| IPVRM w/o ADB/DLW | 1.0 | 40.8 | 11.2 | 12.2 | 12.0 | 15.4 |
| IPVRM w/ ADB/DLW | 1.2 | 40.8 | 13.1 | 12.3 | 13.6 | 16.2 |

**Naive update** (without ADB or DLW), **DLW only**, **ADB only**, and the full **ADB + DLW** strategy. As shown in Figure 6, naive online updating is unstable: RM accuracy quickly collapses toward chance level, and the degraded RM subsequently hurts policy learning, even falling below the frozen-RM baseline. In contrast, both **DLW only** and **ADB only** stabilize training and recover meaningful gains, while the full **ADB + DLW** strategy performs best overall. This shows that online RM refresh is helpful only when it is properly balanced.

We also compare three policy variants on Qwen3-0.6B in Table 8. Replacing Implicit PRM with IPVRM under the same DistRL pipeline improves the average score from 14.7 to 15.4 even without ADB/DLW, and adding ADB/DLW further raises it to 16.2. Together with Figure 6, these results show that stronger prefix-value rewards and balanced online RM updates are complementary in the full system.

## 5. Conclusion

This work studies how to make low-cost implicit process rewards more reliable for verifiable reasoning. We identify a train–inference mismatch in prior implicit PRMs: their sequence-level objectives only constrain an aggregate log-ratio, while inference and RL rely on decomposed local scores for partial reasoning prefixes. To address this mismatch, we propose IPVRM, which redefines implicit supervision as a prefix-conditioned value learning problem. By directly learning the probability of eventual correctness for each prefix and deriving step signals through temporal-difference differences, IPVRM aligns the training target with the prefix-level quantities used for verification and optimization. Empirically, IPVRM improves step-level error localization while preserving sequence-level reranking ability, showing that outcome-only supervision can provide useful process signals when learned as prefix values.

Building on these prefix-level signals, we further introduce Distribution-Level RL (DistRL), which augments trajectory-level policy optimization with updates over high-probability candidate tokens. By using IPVRM-derived TD advantages, DistRL exploits the vocabulary-wide scoring ability of implicit models without requiring additional rollouts. Our results show that distribution-level updates are most effective when supported by reliable prefix values, suggesting prefix-value estimation as a practical foundation for scalable online reward modeling and policy optimization in verifiable reasoning tasks.

## Use of Generative AI Tools

We used large language models as a writing assistant for polishing phrasing and improving readability of the manuscript (and for minor code refactoring such as formatting or commenting). All technical content, claims, experiments, and conclusions were produced and verified by the authors, who take full responsibility for the final manuscript.

## Impact Statement

This paper studies reward modeling and reinforcement learning primarily in verifiable reasoning domains such as mathematics and code, where correctness can be checked by objective evaluators. In these settings, we do not foresee immediate harmful societal impacts beyond those generally associated with improving the performance and efficiency of language models.

However, methods for stronger process supervision and more effective reinforcement learning may also be extended to broader settings. In domains without reliable ground-truth feedback, such methods could exacerbate concerns around over-optimization to imperfect rewards, reward hacking, or the deployment of more capable models in high-stakes applications. We therefore view our framework as most appropriate for objectively verifiable tasks, and believe that broader deployment would require additional care in evaluation, reward design, and human oversight.

## Acknowledgement

Xiaojun Quan acknowledges support from the National Natural Science Foundation of China under Grant No. 62576368.

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

# A. Related Work

**Outcome-Level RL for Reasoning: Progress and Limitations.** Recent progress in mathematical and code reasoning has been largely driven by Reinforcement Learning with Verifiable Rewards (RLVR) (Lambert et al., 2025), where correctness is automatically verified and used as a terminal outcome reward. A representative line of work optimizes reasoning policies directly from such sparse feedback using PPO-style objectives (Yuan et al., 2025b) or variants such as GRPO (Shao et al., 2024). A closely related line of work studies the optimization difficulties induced by sparse outcome rewards, especially insufficient exploration during reasoning. Entropy-based analyses connect performance saturation in reasoning RL to the rapid loss of policy entropy (Cui et al., 2025b), while Zhou et al. (Zhou et al., 2026) show that exploration can be improved by learning sampling temperature from internal states rather than fixing it at inference time. Another line of work instead attempts to densify rewards from the model's own outputs or internal confidence for dense supervision, including majority-vote-based pseudo-rewards in test-time RL (Zuo et al., 2025), self-certainty as an intrinsic reward (Zhao et al., 2025). However, He et al. (He et al., 2026) show that such intrinsic URLVR methods remain fundamentally limited at scale: although they can yield early gains, they largely sharpen the model's initial distribution and often exhibit rise-then-fall behavior when confidence is misaligned with correctness. Taken together, these studies highlight both the progress and the limitations of outcome-level RL for reasoning. Better optimization and exploration can mitigate some challenges of sparse terminal rewards, but sequence-level supervision still offers limited support for fine-grained credit assignment, and purely intrinsic rewards appear insufficient at scale.

**Process Supervision: Dense but Hard to Refresh.** In contrast to intrinsic reward approaches that rely primarily on the model's own outputs or confidence signals, process reward models (PRMs) introduce more grounded external supervision by providing step-level evaluations of intermediate reasoning. This finer-grained supervision supports both inference-time control and policy optimization. A typical explicit PRM learns a scoring function over partial solutions (or step pairs) to predict step correctness or progress from explicit labels or preferences. Early evidence from PRM800K (Lightman et al., 2023) shows that such process supervision can outperform outcome-only signals, but its expert labeling cost has motivated automated and hybrid pipelines. MATH-SHEPHERD (Wang et al., 2024) constructs step rewards by launching multiple rollouts from intermediate states and scoring steps by how often completions reach the correct final answer. AutoPSV (Lu et al., 2024) derives process annotations from an outcome-supervised verifier by tracking relative confidence shifts to identify informative steps, while ACTPRM (Duan et al., 2025) further reduces labeling cost by actively selecting uncertain steps for human or model-based annotation. In parallel, generative PRMs and retrieval-based verification methods use intermediate evidence, such as critiques or informative retrieved context, to support subsequent step scoring or candidate-path verification, thereby improving the semantic basis and interpretability of the resulting judgments (Khalifa et al., 2025; Zhao et al., 2026; Xu et al., 2026). Despite reducing manual annotation, these paradigms remain expensive to refresh: they often require heavy rollout-based label construction or long verifier generations, making it difficult to repeatedly rebuild supervision on behavior-policy rollouts during RL. As the policy distribution shifts, this cost can leave PRMs increasingly stale and misaligned with on-policy data (Gao et al., 2023).

**Implicit process rewards: promise and pitfalls.** Implicit reward models (IM-RMs) parameterize rewards as log-likelihood ratios between a policy and a reference model, as popularized by Direct Preference Optimization (DPO) (Rafailov et al., 2023). This implicit formulation can be decomposed into per-token increments, yielding token-level reward signals in token-level MDPs (Rafailov et al., 2024; Zhong et al., 2025; Gao et al., 2025). Building on the same idea, (Yuan et al., 2025a) show that implicit PRMs can be trained directly from outcome-only labels, and PRIME further supports online PRM updates using behavior-policy rollouts and verifiable outcomes (Cui et al., 2025a), making implicit process rewards a practical bridge between sparse outcome supervision and dense feedback. However, recent work also reports a generalization gap: DPO-induced implicit rewards may generalize worse than explicit reward heads under distribution shifts (Lin et al., 2024), potentially due to heavier reliance on superficial token-level cues and increased sensitivity to perturbations (Razin et al., 2026). These limitations motivate methods that preserve the low-cost and online-update advantages of implicit rewards while improving their faithfulness as process-level supervision.

# B. Discussion on Reference Choice in DistRL

The distribution-level advantage $A_\phi^{\text{TD}}$ is defined through a log-ratio between the reward model $\pi_\phi$ and a reference model $\pi_{\text{ref}}$. We study how reference choice affects learning under two configurations: (i) **SFT Model**, where $\pi_{\text{ref}}$ is fixed to the initial supervised fine-tuned policy $\pi_{\text{sft}}$; and (ii) **Behavior Policy**, where $\pi_{\text{ref}} = \pi_{\text{old}}$, the behavior policy that generated the current batch of rollouts. We provide theoretical and empirical evidence favoring the dynamic reference $\pi_{\text{old}}$ over the static $\pi_{\text{sft}}$.

For a state $s_t$, the soft-value is defined as the log-expected-exponentiated reward under the reference policy:

$$V_\phi(s_t) = \beta \log \mathbb{E}_{y \sim \pi_{\text{ref}}(\cdot | s_t)} \left[ \exp \left( \frac{r_\phi(y)}{\beta} \right) \right]. \tag{20}$$

Expanding this expectation over the next token $y_t$ yields the recursive Bellman consistency equation $V_\phi(s_t) = \beta \log \mathbb{E}_{y_t \sim \pi_{\text{ref}}} [\exp(Q_\phi(s_t, y_t)/\beta)]$. During online reinforcement learning, trajectory samples are generated by the current behavior policy $\pi_{\text{old}}$. Consequently, if we employ a static reference $\pi_{\text{ref}} = \pi_{\text{sft}}$, and let $w_\theta(y_t) = \pi_{\text{sft}}(y_t | s_t)/\pi_{\text{old}}(y_t | s_t)$ be an importance sampling (IS) weight, this value can be estimated as:

$$V_\phi(s_t) = \beta \log \mathbb{E}_{y_t \sim \pi_{\text{old}}} \left[ w_\theta(y_t) \exp \left( \frac{Q_\phi(s_t, y_t)}{\beta} \right) \right]. \tag{21}$$

From this equation, we can see a critical issue of variance. As RL training progresses, $\pi_{\text{old}}$ inevitably deviates from $\pi_{\text{sft}}$, causing the $w_\theta(y_t)$ to increase. In regions where $\pi_{\text{old}}$ explores but $\pi_{\text{sft}}$ has low probability, $w_\theta(y_t)$ approaches zero, while slight overlaps can cause massive

spikes. This variance in $w_\theta(y_t)$ scales the gradients for updating $\pi_\phi$, leading to numerical instability and preventing the reward model from tracking the true value of the current trajectories. Conversely, by setting $\pi_{\text{ref}} = \pi_{\text{old}}$, the importance weight becomes identically 1 ($w_\theta(y_t) \equiv 1$). The estimator matches the sampling distribution:

$$V_\phi(s_t) = \beta \log \mathbb{E}_{y_t \sim \pi_{\text{old}}} \left[ \exp \left( \frac{Q_\phi(s_t, y_t)}{\beta} \right) \right]. \tag{22}$$

This configuration yields an unbiased, minimum-variance estimator, allowing the $\pi_\phi$ to stably learn the value landscape of the exploration region without suffering from distribution mismatch.

Empirical results corroborate this analysis. We compared the two configurations across multiple mathematical reasoning benchmarks using Qwen3-0.6B and Llama3.2-1B base models. As shown in Table 9, the **Behavior Policy** reference consistently outperforms the **SFT Model**. For Qwen3-0.6B, using the behavior policy reference improves the average accuracy from 15.2% to 16.7%, with similar gains observed for Llama3.2-1B (3.8% to 4.8%). These results confirm that the improved distributional alignment and reduced variance provided by the dynamic reference translate directly into more effective policy optimization.

*Table 9.* Impact of reference choice on distribution-level RL across mathematical benchmarks (AVG@8).

| Base Model | Reference Choice | AIME | MATH 500 | Minerva Math | Olympiad Bench | AMC | Avg. |
|---|---|---|---|---|---|---|---|
| **Qwen3-0.6B** base | SFT Model | 1.4 | 39.8 | 12.1 | 11.3 | 11.3 | 15.2 |
| | Behavior Policy | **1.5** | **42.5** | **13.3** | **12.1** | **14.0** | **16.7** |
| **Llama3.2-1B** base | SFT Model | **0.3** | 10.3 | 2.9 | **3.5** | 2.1 | 3.8 |
| | Behavior Policy | 0.1 | **13.2** | **3.4** | 3.1 | **4.2** | **4.8** |

# C. Implementation Details

**Supervised Fine-Tuning**   All models were fine-tuned on the PRIME-RL/Eurus-2-SFT-Data dataset for one epoch, using a global batch size of 64, a warmup ratio of 0.01, and a sequence cutoff length of 3072. For the Qwen3-8B-base model, we performed LoRA fine-tuning with rank 128, LoRA scaling factor 128, and a learning rate of $1 \times 10^{-4}$; the LoRA adapters were merged before subsequent RM and RL training. All other models were trained with a learning rate of $5 \times 10^{-5}$. The SFT phase was conducted using the LLaMA-Factory[4] framework.

**Reward Model**   All reward models were initialized from the corresponding supervised fine-tuning (SFT) checkpoints, with training conducted using a learning rate of $5 \times 10^{-7}$ and a global batch size of 64. For DPO-RM and Implicit PRM, we set $\beta = 0.05$, following the recommendation in the official implementation of Implicit PRM[5]. For QRM, we adopted $\beta = 0.2$ and $m = 2.0$ as suggested in its original paper. For our proposed IPVRM, we used $\beta = 10$ and $m = 5.0$. The training of all reward models was implemented based on LLaMA-Factory. For EndoRM, we leveraged SFT-tuned models to provide token-level rewards and set the decay factor to 0.95, as specified in the appendix of its original paper.

**Reinforcement Learning**   Following the official documentation[6], we used LoRA with rank 128 and a learning rate of $3 \times 10^{-5}$ to match the convergence behavior of full-parameter training while reducing GPU memory consumption. To stabilize RL optimization, we sampled $n = 4$ responses per prompt and used an oversampling factor of 2: specifically, we sampled 512 prompts per batch and selected a final batch of 256 prompts exhibiting mixed correctness. For reward models that support online updates—namely DPO-RM, Implicit-PRM, and IPVRM—we applied a learning rate of $10^{-4}$ using rollouts from the behavior policy together with rule-based outcome labels. Within the DistRL framework, we set $P_{\min} = 0.1$ for candidate selection and chose $\alpha = 0.1$ to balance the advantage signals $A_\phi^{\text{TD}}$ and $A_\phi^{\text{GAE}}$, tuning $\alpha$ on the validation set over $\{0.05, 0.1, 0.2, 0.5\}$. For PPO clipping, we followed (Yu et al., 2025) and set $\varepsilon_{\text{low}} = 0.20$ and $\varepsilon_{\text{high}} = 0.28$. RL training was implemented based on VeRL PRIME[7]. The dataloader seed was set to 42. No other seeds were configured, as vLLM's strong non-determinism makes them meaningless[8].

**Minibatch-normalized TD advantages.**   The normalization below is applied only to stabilize PPO-style optimization. It rescales the TD signal from Eq. (10) and does not define a different value function. In practice, we compute one-step TD advantages in Eq. (10) using a minibatch-normalized value function to stabilize the scale of the TD difference. Let $\mathcal{B}$ denote the set of all prefix states whose values are evaluated in the current PPO update (across all sampled rollouts and timesteps). We compute

$$\mu_V = \text{mean}_{s \in \mathcal{B}}[V_\phi(s)], \qquad \sigma_V = \text{std}_{s \in \mathcal{B}}[V_\phi(s)], \tag{23}$$

---

[4] https://github.com/hiyouga/LLaMA-Factory
[5] https://github.com/PRIME-RL/ImplicitPRM/tree/main/train/tasks
[6] https://verl.readthedocs.io/en/latest/advance/ppo_lora.html
[7] https://github.com/volcengine/verl/tree/main/recipe/prime
[8] https://github.com/volcengine/verl/issues/1683

and define the normalized value

$$\widetilde{V}_\phi(s) \;=\; \frac{V_\phi(s) - \mu_V}{\sigma_V + \epsilon}, \tag{24}$$

where $\epsilon$ is a small constant for numerical stability. We then form the TD advantage by replacing $V_\phi$ with $\widetilde{V}_\phi$ in Equation (10):

$$A_\phi^{\mathrm{TD}}(s_t, y_t') \;=\; \widetilde{V}_\phi(s_t \oplus y_t') \;-\; \widetilde{V}_\phi(s_t) = \beta \log \frac{\pi_\phi(y_t' \mid s_t)}{\pi_{\mathrm{old}}(y_t' \mid s_t)} \Big/ (\sigma_V + \epsilon). \tag{25}$$

The normalization statistics $(\mu_V, \sigma_V)$ are computed as minibatch moments (without backpropagating through them).

**Prompt-group normalization before GAE.** Again, this prompt-group normalization is applied only to the optimizer inputs used by GAE; it is not part of the reward-model definition itself. For sequence-level advantages, we compute $A_\phi^{\mathrm{GAE}}$ Equation (8) from *prompt-group normalized* TD advantages. For each prompt $x$, we draw $n$ rollouts $\{y^{(k)}\}_{k=1}^n \sim \pi_{\mathrm{old}}(\cdot \mid x)$ (as in Equation (13)), and define the corresponding prompt group

$$\mathcal{G}(x) \;=\; \{(k, t) \,:\, k \in \{1, \ldots, n\}, \; t \in \{1, \ldots, T^{(k)}\}\}, \tag{26}$$

where $T^{(k)}$ is the length of rollout $y^{(k)}$. Let $s_t^{(k)} = (x, y_{<t}^{(k)})$ and let $\delta_t^{(k)} = A_\phi^{\mathrm{TD}}(s_t^{(k)}, y_t^{(k)})$ denote the token-level TD advantage for the sampled token, computed as the sampled-token special case of Eq. (25).

We compute prompt-group moments over $\mathcal{G}(x)$:

$$\mu_{\mathrm{TD}}(x) \;=\; \mathrm{mean}_{(k,t) \in \mathcal{G}(x)}\left[\delta_t^{(k)}\right], \qquad \sigma_{\mathrm{TD}}(x) \;=\; \mathrm{std}_{(k,t) \in \mathcal{G}(x)}\left[\delta_t^{(k)}\right], \tag{27}$$

and normalize TD advantages within the prompt group:

$$\widehat{\delta}_t^{(k)} \;=\; \frac{\delta_t^{(k)} - \mu_{\mathrm{TD}}(x)}{\sigma_{\mathrm{TD}}(x) + \epsilon}. \tag{28}$$

Finally, we apply the standard GAE aggregation Equation (8) using the normalized TD advantages:

$$A_\phi^{\mathrm{GAE}}(s_t^{(k)}, y_t^{(k)}) \;=\; \sum_{i=t}^{T^{(k)}} (\gamma\lambda)^{i-t} \, \widehat{\delta}_i^{(k)}. \tag{29}$$

As above, the group statistics $(\mu_{\mathrm{TD}}(x), \sigma_{\mathrm{TD}}(x))$ are treated as moment estimates and are not differentiated through.

# D. Sensitivity to $m$ and $P_{\min}$

We report sensitivity analyses for the two hyperparameters that most directly control IPVRM training and DistRL candidate selection: the margin $m$ in the IPVRM objective and the probability threshold $P_{\min}$ for the candidate set. Unless otherwise specified, all results below are based on Qwen3-0.6B.

**Sensitivity to the margin $m$.** We vary the margin $m \in \{0, 5, 10\}$ and report both BoN average accuracy and PROCESSBENCH average $F_1$.

*Table 10.* Sensitivity to the margin $m$ on Qwen3-0.6B.

| $m$ | BoN-AVG | PB-AVG |
|---|---|---|
| 0 | 26.1 | 37.0 |
| 5 | 28.1 | 39.1 |
| 10 | 27.5 | 39.1 |

These results show that the method is stable around the default choice $m = 5$. Moving from $m = 5$ to $m = 10$ changes performance only slightly, while removing the margin ($m = 0$) causes a clearer degradation, especially on BoN.

**Sensitivity to the candidate threshold $P_{\min}$.** We sample 1,000 training prompts, roll out the Qwen3-0.6B SFT policy, and record the full-vocabulary next-token probabilities at each step. From these distributions, we compute the average candidate set size and the retained policy probability mass for different values of $P_{\min}$.

These results make $P_{\min} = 0.1$ a well-justified operating point: it keeps the candidate set very small in practice while still retaining most of the policy mass. Lower thresholds yield only limited additional coverage, whereas larger thresholds start to discard nontrivial policy mass.

*Table 11.* Sensitivity to the candidate threshold $P_{\min}$ on Qwen3-0.6B.

| $P_{\min}$ | Avg Candidate Size | Prob. Mass Coverage |
|---|---|---|
| 0.05 | 1.54 | 97.57% |
| 0.10 | 1.33 | 96.11% |
| 0.20 | 1.16 | 93.58% |

*Table 12.* Best-of-$N$ sampling accuracy (%) on AMC-23 with SFT models.

*(a)* Qwen3-0.6B Base

| BON | Q-RM | EndoRM | DPO-RM | Implicit PRM | IPVRM |
|---|---|---|---|---|---|
| @4 | **25.0** | 15.0 | 22.5 | **25.0** | **25.0** |
| @16 | 17.5 | 12.5 | 22.5 | 17.5 | **27.5** |
| @64 | **27.5** | 20.0 | 25.0 | 17.5 | 22.5 |
| AVG | 23.3 | 15.8 | 23.3 | 20.0 | **25.0** |

*(b)* Qwen3-8B Base

| BON | Q-RM | EndoRM | DPO-RM | Implicit PRM | IPVRM |
|---|---|---|---|---|---|
| @4 | **50.0** | 25.0 | **50.0** | 47.5 | 47.5 |
| @16 | 42.5 | 30.0 | 42.5 | 42.5 | **50.0** |
| @64 | **45.0** | 25.0 | 42.5 | 42.5 | **45.0** |
| AVG | 45.8 | 26.7 | 45.0 | 44.2 | **47.5** |

*(c)* Llama3.2-1B

| BON | Q-RM | EndoRM | DPO-RM | Implicit PRM | IPVRM |
|---|---|---|---|---|---|
| @4 | **5.0** | 2.5 | **5.0** | **5.0** | 5.0 |
| @16 | 2.5 | 2.5 | 2.5 | **5.0** | **5.0** |
| @64 | 5.0 | **10.0** | **10.0** | **10.0** | **10.0** |
| AVG | 4.2 | 5.0 | 5.8 | **6.7** | **6.7** |

*(d)* Llama3.2-3B

| BON | Q-RM | EndoRM | DPO-RM | Implicit PRM | IPVRM |
|---|---|---|---|---|---|
| @4 | **10.0** | 7.5 | **10.0** | **10.0** | **10.0** |
| @16 | **7.5** | **7.5** | **7.5** | **7.5** | 5.0 |
| @64 | 7.5 | 12.5 | 10.0 | 10.0 | **17.5** |
| AVG | 8.3 | 9.2 | 9.2 | 9.2 | **10.8** |

# E. Complete BON test results on AMC-23

We additionally report Best-of-$N$ results on **AMC-23** in Table 12, following the same evaluation protocol as Table 1.

# F. Limitations and Discussion

While our framework demonstrates significant improvements in reasoning tasks, we identify several limitations that clarify the scope of our work and point toward future research directions.

**Outcome-Induced Label Noise vs. Supervision Density** IPVRM supervises *every* prefix using only a terminal outcome label, which may introduce label noise: a prefix can contain early incorrect or weak reasoning steps, yet the trajectory may later recover and still reach the correct final answer, causing such prefixes to be labeled as "correct." This reflects a fundamental trade-off in outcome-based supervision. Rather than modeling step-level correctness explicitly, IPVRM is trained to estimate a prefix-conditioned value $V(s_t)$, namely the expected probability of eventual success from the current state. Outcome-derived supervision therefore sacrifices some label fidelity in exchange for much denser supervision, yielding gradients at every token. Empirically, this trade-off is effective: despite the imperfect supervision, IPVRM produces stronger process-level discrimination after training and achieves improved error localization on PROCESSBENCH compared to prior implicit reward models.

**Scope of Verifiable Reward Environments** Our current framework is designed for reasoning tasks with *verifiable* outcomes, such as mathematics and code, where correctness can be determined by an objective and automated verifier. This setting enables reliable large-scale supervision and supports frequent online updates of the reward model during RL. However, this reliance on verifiable rewards also limits the applicability of our approach. Extending the framework to open-ended or subjective domains (e.g., creative writing) is not straightforward, as such settings lack a consistent source of ground-truth feedback. In these cases, one would need to rely on frozen reward models or incorporate costly human-in-the-loop evaluation, which we do not explore in this work. Therefore, all empirical results in this paper should be interpreted within the scope of verifiable reasoning tasks, rather than as evidence of effectiveness in arbitrary open-ended generation settings.

**Computational Efficiency** DistRL introduces additional computation through the use of one-step TD advantages $A_\phi^{\text{TD}}$ over candidate tokens. However, this overhead is limited in practice due to the design of IPVRM. Unlike explicit PRMs that often require separate value-head forward passes or additional Monte Carlo rollouts for each candidate action, IPVRM fully reuses the generative head of the language model. At each timestep $t$, a single forward pass produces the full logit distribution over the vocabulary, which already contains all information needed to compute both sequence-level GAE advantages for the sampled token and one-step TD advantages for all candidates in $\mathcal{C}$. As a result, DistRL densifies the learning signal without introducing additional forward passes. Table 13 reports the peak GPU memory usage and average wall-clock time per training step for IPVRM+DistRL compared with IPVRM+GRPO. Across different model scales, DistRL incurs only a modest increase in memory, confirming that the proposed distribution-level supervision

improves sample efficiency while maintaining comparable computational cost.

*Table 13.* Peak GPU memory usage and average wall-clock time per training step for IPVRM with DistRL and GRPO.

| Model | Method | Peak Memory (GB) | Avg Time / Step (s) |
|---|---|---|---|
| Qwen3-0.6B | GRPO | 30.20 | $487.5 \pm 27.1$ |
| | DistRL | 30.88 | $458.7 \pm 33.2$ |
| Llama3.2-1B | GRPO | 31.30 | $349.8 \pm 12.6$ |
| | DistRL | 31.52 | $319.6 \pm 13.0$ |

## G. ProcessBench Evaluation under Different Scoring Protocols

In our paper, the ProcessBench evaluation of implicit reward models follows a slightly different scoring procedure from prior work. Specifically, existing implementations (Yuan et al., 2025a) interpret the sigmoid of the *prefix* log-likelihood ratio,

$$\sigma\Big( \beta \sum_{i=1}^{t_1} \log \frac{\pi_\phi(y_i|s_i)}{\pi_{\text{ref}}(y_i|s_i)} \Big), \tag{30}$$

as the probability that a reasoning step is correct, where $t_1$ is the index of the last token of the current process. In contrast, we adopt a *process-level* formulation that scores only the tokens within the current reasoning step:

$$\sigma\Big( \beta \sum_{i=t_2}^{t_1} \log \frac{\pi_\phi(y_i|s_i)}{\pi_{\text{ref}}(y_i|s_i)} \Big), \tag{31}$$

where $t_2$ is the index of the first token of this process. This formulation isolates the contribution of the current step, rather than accumulating signals from earlier parts of the trajectory.

Both formulations first compute a log-likelihood ratio and then map it to a probability via a sigmoid function; the only difference lies in whether the aggregation is taken over the full prefix or restricted to the current step. For completeness, we report PROCESSBENCH results under both the standard prefix-based protocol of (Yuan et al., 2025a) and our process-level variant.

Table 14 reports $F_1$ scores across GSM8K, MATH, OlympiadBench, and OmniMath. IPVRM consistently outperforms DPO-RM and Implicit PRM under both formulations, demonstrating that its advantage is not dependent on the choice of evaluation protocol. At the same time, the process-level formulation yields consistently higher $F_1$ scores across all methods, suggesting that isolating the current reasoning step provides a more faithful signal for error localization. This is consistent with the design goal of ProcessBench, which focuses on identifying incorrect intermediate steps.

*Table 14.* ProcessBench $F_1$ under different evaluation protocols.

| Protocol | Method | GSM8K | MATH | Olympiad | OmniMath | Avg |
|---|---|---|---|---|---|---|
| Prefix-based | DPO-RM | 27.0 | 27.4 | 19.8 | 18.4 | 23.2 |
| | Implicit PRM | 30.4 | 28.2 | 19.9 | 16.9 | 23.9 |
| | IPVRM (Ours) | 42.6 | 36.4 | 24.3 | 25.2 | 32.1 |
| Process-level | DPO-RM | 33.1 | 32.7 | 23.5 | 22.9 | 28.1 |
| | Implicit PRM | 30.3 | 31.0 | 21.5 | 18.8 | 25.4 |
| | IPVRM (Ours) | 52.8 | 42.1 | 30.0 | 31.3 | 39.1 |

