# OpenReview forum: "Unleashing Implicit Rewards: Prefix-Value Learning for Distribution-Level Optimization"
_ICML.cc/2026/Conference — ICML 2026 regular_

### Official Review · Reviewer_jDZC · 2026-03-11

**Soundness:** 3
**Presentation:** 4
**Significance:** 3
**Originality:** 3
**Overall Recommendation:** 4
**Confidence:** 3

**Summary:**

This paper studies implicit process reward modeling for RL-based reasoning. They paper proposes the Implicit Prefix-Value Reward Model (IPVRM), which directly learns a prefix-conditioned value estimating the probability that the current prefix will eventually lead to a correct final answer. Building on this reward model, the paper further proposes Distribution-Level RL (DistRL), which extends RL updates beyond sampled tokens to high-probability candidate tokens in the next-token distribution, with the goal of extracting denser learning signal without extra rollouts.
The empirical study covers several model sizes (Qwen3-0.6B/8B and Llama3.2-1B/3B), reward-model evaluation on Best-of-N selection and PROCESSBENCH, downstream policy evaluation on multiple math reasoning benchmarks, and ablations on temporal weighting, online reward-model updating, and reference-policy choices.

**Compliance With Llm Reviewing Policy:**

Affirmed.

**Final Justification:**

My concerns have been adequately addressed. I wiil keep my score.

**Key Questions For Authors:**

1. The paper states that it evaluates IM-RMs using a process log-likelihood ratio variant rather than the prefix-based formulation used in prior evaluation code because it gives higher F1 across methods. Please report the main PROCESSBENCH comparisons under both protocols. If IPVRM remains clearly stronger under the more standard setup, that would strengthen my confidence in the main claim.

2. Since some of the final policy improvements from DistRL over GRPO are relatively modest, can the authors provide repeated-run statistics, confidence intervals, or any other evidence that these gains are robust and not due to training variance?

**Limitations:**

yes

**Strengths And Weaknesses:**

**Strengths**

1. The move from decomposed implicit rewards to prefix-conditioned values is intuitive and reasonably well aligned with the inference-time query. The proposed TD-difference construction is simple, easy to implement, and conceptually cleaner than asking a sequence-level objective to implicitly recover faithful token-level credit.

2. The paper evaluates across four backbones and multiple reward-model and downstream-policy benchmarks. The PROCESSBENCH gains are particularly strong and consistent, which is important because that benchmark directly targets the central claim about process-level reliability under format shift. The downstream RL results also generally favor the proposed combination of IPVRM + DistRL.

**Weaknesses**

1. The reward-model gains on PROCESSBENCH are striking, but the final policy improvements from DistRL over GRPO with the same IPVRM are smaller. This does not negate the contribution, but it does limit impact. In other words, the reward-model story is currently stronger than the final policy-optimization story.

---

> ### Author Rebuttal · Authors · 2026-03-29
>
> # Response to Reviewer jDZC
>
> Thank you for the constructive feedback. We directly followed your two main suggestions with additional experiments.
>
> ### Q1: Please report the main ProcessBench comparisons under both the standard prefix-based protocol and your process variant.
>
> **Response:**
> We evaluated both the standard **prefix-based formulation** and our **process log-likelihood ratio variant**, and IPVRM remains clearly stronger under both.
>
> The standard prefix-based formulation accumulates log-ratios from the beginning of the prompt, which can introduce historical bias from earlier steps. Our process variant isolates the current step by scoring only the tokens generated within that step. To directly address your concern, we report the results under both protocols below.
>
> We therefore re-evaluated the **Qwen3-0.6B** reward models on ProcessBench under both protocols:
>
> | Eval Protocol | Method | GSM8K | MATH | OlympiadBench | OmniMATH | Avg |
> |---|---|---:|---:|---:|---:|---:|
> | Standard (Prefix-based) | DPO-RM | 27.0 | 27.4 | 19.8 | 18.4 | 23.2 |
> | Standard (Prefix-based) | Implicit PRM | 30.4 | 28.2 | 19.9 | 16.9 | 23.9 |
> | Standard (Prefix-based) | **IPVRM (Ours)** | **42.6** | **36.4** | **24.3** | **25.2** | **32.1** |
> | Process Variant (Ours) | DPO-RM | 33.1 | 32.7 | 23.5 | 22.9 | 27.8 |
> | Process Variant (Ours) | Implicit PRM | 30.3 | 31.0 | 21.5 | 18.8 | 25.4 |
> | Process Variant (Ours) | **IPVRM (Ours)** | **52.8** | **42.1** | **30.0** | **31.3** | **39.1** |
>
> These results make the main conclusion unambiguous: **IPVRM remains clearly stronger than both DPO-RM and Implicit PRM under both protocols.** The main conclusion therefore does **not** depend on the evaluation variant.
>
> ---
>
> ### Q2: Since the final policy improvements from DistRL over GRPO are relatively modest, can you provide repeated-run statistics or other evidence that these gains are robust?
>
> **Response:**
> Yes. We repeated the **Qwen3-0.6B** comparison between **DistRL + IPVRM** and **GRPO + IPVRM** over **3 seeds**:
>
> | Method | Seed 1 | Seed 2 | Seed 3 | Mean ± Std |
> |---|---:|---:|---:|---:|
> | DistRL + IPVRM | 16.12 | 15.58 | 16.82 | **16.17 ± 0.62** |
> | GRPO + IPVRM | 15.48 | 16.24 | 15.28 | **15.67 ± 0.51** |
>
> This yields a **+0.51 AVG@8** improvement for DistRL on average across 3 seeds. The repeated runs make clear that this gain is **reproducible rather than seed-dependent**. This directly supports the intended framing of the paper: **IPVRM is the main contribution, and DistRL is the optimization component that consistently extracts additional performance from the learned reward model.**

---

> > ### Author Rebuttal · Reviewer_jDZC · 2026-04-02
> >
> > My concerns have been adequately addressed. I wiil keep my score.

---

### Official Review · Reviewer_WY8J · 2026-03-15

**Soundness:** 3
**Presentation:** 3
**Significance:** 2
**Originality:** 3
**Overall Recommendation:** 4
**Confidence:** 3

**Summary:**

The paper claims that standard implicit PRMs are trained with a sequence-level outcome objective but are later used as local step scorers, which creates a train-inference mismatch. To address this, the authors propose IPVRM, which learns prefix-conditioned values from outcome labels at every prefix, and DistRL, which adds PPO-style updates not only on sampled tokens but also on high-probability candidate tokens. The reported results show large gains for process verification on PROCESSBENCH and smaller but mostly positive gains for downstream RL across Qwen and Llama backbones.

**Compliance With Llm Reviewing Policy:**

Affirmed.

**Final Justification:**

The authors addressed my two questions and clarified the justification, so I have increased my confidence score to 3.

**Key Questions For Authors:**

Please check the Weaknesses part.

**Strengths And Weaknesses:**

## Strengths

1. The paper is well motivated. The gap between cheap outcome labels and costly step labels is real, and the paper gives a clear empirical story that sequence ranking and process verification are not the same thing. In particular, Table 1 and Table 2 support the paper’s main point: methods can look competitive on Best-of-N reranking while still failing to localize reasoning errors under format shift.


2. The PROCESSBENCH result is good. IPVRM improves average F1 quite a lot over DPO-RM and Implicit PRM, especially on smaller models. For example, on Qwen3-0.6B, the average F1 rises from 27.8 with DPO-RM and 25.4 with Implicit PRM to 39.1 with IPVRM. That is a meaningful jump, and it is more convincing than the Best-of-N gains, which are smaller.

3. The paper is also easy to follow at a high level. Figures 1, 3, and 4 do a good job of showing the method and the claimed tradeoffs.

## Weaknesses

1. My main concern is technical clarity and possibly soundness around the definition of the “value” used by IPVRM and DistRL. Equation 9 trains a length-normalized cumulative log-ratio with a sigmoid and margin. The text then says that the model learns a prefix-conditioned value estimating the probability of eventual correctness. But Equation 10 writes the TD advantage as an exact value difference that equals a single-token log-ratio, and Appendix C adds minibatch normalization on top of that. Is this object the same?

2. The training target is still outcome-derived at every prefix, not local step correctness. A trajectory can contain early weak or wrong steps and later recover, yet all prefixes on that trajectory are trained toward the positive label. The appendix does acknowledge this limitation, but the main text sometimes sounds stronger than what the method directly gives.

---

> ### Author Rebuttal · Authors · 2026-03-29
>
> # Response to Reviewer WY8J
>
> Thank you for the careful reading and for pinpointing two central questions about the method.
>
> ### Q1: Is the “value” used in Eq. (9), Eq. (10), and Appendix C actually the same object?
>
> **Response:**
> Yes, the key point is that **Eq. (9), Eq. (10), and Appendix C are built from the same underlying prefix-value / cumulative log-ratio representation**, while using **different scalings or transforms** for different optimization roles.
>
> The shared underlying representation is the cumulative prefix score
> $V_\phi(s_t) = \beta \sum_{i=1}^t \log \frac{\pi_\phi(y_i \mid s_i)}{\pi_{\text{ref}}(y_i \mid s_i)}.$
>
> Eq. (9), Eq. (10), and Appendix C all derive from this same object, but they use it in different forms because the mathematical requirements of reward-model training and RL optimization are fundamentally different.
>
> - **Eq. (9) (RM training):** The reward model is trained with a sigmoid BCE objective. In this phase, directly applying BCE to the raw cumulative score would become increasingly ill-scaled as the prefix grows. We therefore use a **length-normalized** form to keep the reward-model objective well behaved across different prefix lengths.
> - **Eq. (10) (TD signal in RL):** In the RL phase, we return to the **unnormalized cumulative value** when forming the TD signal. This is the natural form for TD differences because it preserves the clean token-level decomposition between consecutive prefixes. If the length normalization were kept here, the TD difference would introduce extra cross-terms and no longer align cleanly with the token-level log-ratio increment.
> - **Appendix C (optimization stability):** The minibatch normalization there is a further **implementation-level rescaling** applied after the TD signal is formed, solely to stabilize PPO-style optimization. It does not change the underlying quantity being modeled.
>
> So the correct interpretation is: the method is built around a single underlying prefix-value representation, but Eq. (9), Eq. (10), and Appendix C use different transformed versions of it for different optimization purposes. We will make this relationship explicit in the revision.
>
> ---
>
> ### Q2: The training target is still outcome-derived at every prefix, not local step correctness. A trajectory can contain early weak or wrong steps and later recover, yet all prefixes on that trajectory are trained toward the positive label. The appendix does acknowledge this limitation, but the main text sometimes sounds stronger than what the method directly gives.
>
> **Response:**
> This observation is correct, and the key point is to state the modeling target precisely. IPVRM’s target is a **prefix-conditioned future-success value**: the probability that the current prefix will eventually lead to a correct final outcome.
>
> Under this objective, outcome-derived supervision is the appropriate scalable signal. It is true that a trajectory may contain an early weak or incorrect step and later recover, so a positive terminal outcome is not a perfect label for whether each intermediate step is locally correct in isolation. But that is **not** the quantity IPVRM is designed to estimate. IPVRM is instead learning whether the **current prefix as a whole** still lies on a path with high expected future success. We will make this more explicit in the main text.
>
> The relevant trade-off is therefore between **label fidelity for local step validity** and **dense, scalable supervision at every prefix**. Outcome-derived prefix supervision is imperfect for judging local step correctness token by token, but it provides supervision at **every prefix**, is directly available from verifiable rewards, and makes frequent online reward-model updates feasible. This is exactly the regime targeted by our method.
>
> Empirically, this modeling choice is effective: although IPVRM is trained only from outcome supervision, it yields substantially stronger process discrimination on ProcessBench and provides a more useful optimization signal in downstream RL than prior implicit reward formulations. So while outcome-derived prefix supervision is not equivalent to ground-truth local step labels, it is still an effective and scalable training signal for the objective IPVRM is designed to solve.

---

> > ### Author Rebuttal · Reviewer_WY8J · 2026-04-03
> >
> > Thanks for your explanations. I would like to keep my score.

---

### Official Review · Reviewer_4Qnz · 2026-03-15

**Soundness:** 3
**Presentation:** 3
**Significance:** 3
**Originality:** 3
**Overall Recommendation:** 4
**Confidence:** 3

**Summary:**

This paper identifies and addresses two interrelated problems in the use of implicit process reward models (PRMs) for reinforcement learning with verifiable rewards (RLVR). First, the authors observe that standard implicit PRMs (as used in PRIME) suffer from a train-inference mismatch: training fits a sequence-level objective to terminal correctness labels, yet inference queries prefix-level scores for step-wise credit assignment. To resolve this, the paper introduces the Implicit Prefix-Value Reward Model (IPVRM), which directly learns a prefix-conditioned state-value function $V_\phi(s_t) = P(\text{correct} \mid \text{prefix})$ using a binary cross-entropy loss with a learnable margin. This reformulation ensures that the training objective mirrors the inference query, yielding better-calibrated process-level scores. Second, the authors propose Distribution-Level RL (DistRL), a dual-branch policy optimization framework. Beyond the standard sampled-token branch (which uses GAE advantages), DistRL introduces a candidate-token branch that computes one-step TD advantages for high-probability unsampled tokens across the vocabulary, enabling dense counterfactual updates without additional rollouts. The framework is completed by an online reward model update strategy featuring Adaptive Difficulty Boundary (ADB) and Dynamic Loss Weighting (DLW) to handle prompt-difficulty heterogeneity and label imbalance. Experiments are conducted on Qwen3-0.6B/8B-Base and Llama-3.2-1B/3B across ProcessBench, Best-of-N selection, and five downstream mathematical reasoning benchmarks, demonstrating consistent improvements over DPO-RM, Implicit PRM, GRPO, and PRIME baselines.

**Compliance With Llm Reviewing Policy:**

Affirmed.

**Key Questions For Authors:**

1. **TD approximation quality.** Can you compare $A^{\text{TD}}(s_t, y'_t)$ against Monte Carlo returns from actual rollouts starting at $s_t \oplus y'_t$? This would clarify whether candidate-token gains come from accurate credit assignment or regularization.

2. **Statistical significance of DistRL gains.** Can you report standard deviations for Table 3? If variance exceeds the +0.1--1.4 point improvements, the primary contribution would be IPVRM rather than DistRL.

3. **Sensitivity to m and $P_{\min}$.** These hyperparameters ($m=5.0$, $P_{\min}=0.1$) critically control IPVRM calibration and candidate set size but lack sensitivity analysis.

**Limitations:**

The authors discuss limitations in Appendix E, covering three aspects: (1) outcome-induced label noise, where correct trajectories may contain intermediate errors that the prefix-level supervision cannot distinguish from genuinely correct reasoning; (2) the restriction to verifiable reward environments with binary outcome labels; and (3) computational overhead of DistRL. These are relevant and honestly presented. However, the discussion could be more thorough regarding: (a) the potential for the candidate-token branch to amplify reward model biases (if $V_\phi$ systematically misevaluates certain token types, the distribution-level update would propagate this error across many tokens per step); (b) the scalability to longer reasoning traces where the one-step TD approximation may become less reliable; and (c) broader societal impacts are dismissed in a single sentence ("none of which we feel must be specifically highlighted here"), which, while probably accurate for a math-focused method, could be more carefully considered given that the framework is presented as general-purpose.

**Strengths And Weaknesses:**

### Strengths

1. **Principled resolution of the train-inference mismatch.** Directly training $V_\phi(s_t)$ to predict $P(\text{correct} \mid \text{prefix})$ via per-prefix BCE is conceptually clean and yields compelling gains (+11.3 avg F1 over DPO-RM on ProcessBench).

2. **Novel distribution-level optimization.** The candidate-token branch extracts dense TD-advantage signals for high-probability unsampled tokens from a single rollout, a meaningful departure from trajectory-only RL that shows consistent gains in Table 3.

3. **Comprehensive evaluation and ablations.** Three evaluation axes (RM quality, inference scaling, policy optimization), four models across two families, and well-designed ablations (temporal weighting, online RM, reference choice) effectively isolate each component's contribution.

### Weaknesses

1. **Fragile one-step TD approximation for candidate tokens.** Trajectories after an unsampled token may diverge drastically from the one-step lookahead assumption, yet no comparison with full-rollout Monte Carlo returns is provided.

2. **Modest DistRL gains without statistical validation.** Improvements over GRPO w/ IPVRM are often +0.1--1.4 points (Table 3) with no standard deviations or confidence intervals reported, leaving statistical significance unclear.

3. **Math-only evaluation.** Despite the general-sounding title, all experiments use math benchmarks with deterministic verifiers; code generation, theorem proving, and non-binary-reward settings are unexplored.

---

> ### Author Rebuttal · Authors · 2026-03-29
>
> # Response to Reviewer 4Qnz
>
> Thank you for the detailed review. We followed your suggestions with additional diagnostics and ablations.
>
> ### Q1: TD approximation quality. Can you compare $A^{TD}(s_t, y'_t)$ against Monte Carlo returns from actual rollouts starting at $s_t \oplus y'_t$? This would clarify whether candidate-token gains come from accurate credit assignment or regularization.
>
> **Response:**
> Yes. We tested on **MATH-500** using the **Qwen3-0.6B SFT policy + IPVRM**. We randomly selected one truncation point per trajectory, took the **top-5 candidate tokens**, and compared the one-step TD signal $A^{TD}(s_t, y'_t)$ with the Monte Carlo outcome obtained by rolling out from $s_t \oplus y'_t$ to completion.
>
> Across **2,500 candidate branches**, the **absolute prefix value** remains predictive of final correctness (**AUC-ROC = 0.64**), while the correlation between one-step $A^{TD}$ and single-rollout binary outcomes is weak but still positive (**average Pearson = 0.0226**).
>
> This is expected in long-horizon reasoning, where final success is rarely determined by a single next token. Thus, $A^{TD}$ is not an exact Monte Carlo credit signal, but a **useful local optimization signal**: it does not perfectly predict long-horizon return, yet still provides cheap, reward-aligned updates over plausible candidate tokens without extra rollouts.
>
> To determine whether the candidate-token gains are merely a regularization effect or instead reflect genuinely better use of the learned RM, we directly examined training-time optimization by recording the mean per-step RM score during training,
> $\frac{1}{|y|}\left(\log \pi_\phi(y\mid x)-\log \pi_{\text{ref}}(y\mid x)\right),$
> which measures how strongly policy outputs are favored over the reference policy.
>
> Across all four backbones, DistRL, which uses $A^{TD}$, consistently achieves a higher mean per-step RM score than GRPO, directly showing that it makes more effective use of the learned reward signal:
> - **Qwen3-0.6B**: $-0.140$ vs. $-0.181$
> - **Qwen3-8B**: $-0.178$ vs. $-0.202$
> - **Llama-1B**: $-0.040$ vs. $-0.050$
> - **Llama-3B**: $-0.024$ vs. $-0.025$
>
> We also measured **rollout verifier accuracy** during training, and DistRL is again higher than GRPO on all four backbones:
> - **Qwen3-0.6B**: 23.5 vs. 23.0
> - **Qwen3-8B**: 41.6 vs. 41.4
> - **Llama-1B**: 7.7 vs. 7.4
> - **Llama-3B**: 15.9 vs. 15.5
>
> Taken together, these results clarify the role of TD: **$A^{TD}$ is not an exact Monte Carlo credit signal, but a practically useful local optimization signal** that consistently improves both RM score and rollout correctness during training.
>
> ---
>
> ### Q2: Statistical significance of DistRL gains. Can you report standard deviations / repeated-run statistics?
>
> **Response:**
> Yes. We repeated the **Qwen3-0.6B** comparison between **DistRL + IPVRM** and **GRPO + IPVRM** over **3 seeds**:
>
> | Method | Seed 1 | Seed 2 | Seed 3 | Mean ± Std |
> |---|---:|---:|---:|---:|
> | DistRL + IPVRM | 16.12 | 15.58 | 16.82 | **16.17 ± 0.62** |
> | GRPO + IPVRM | 15.48 | 16.24 | 15.28 | **15.67 ± 0.51** |
>
> This yields a **+0.51 AVG@8** gain for DistRL and makes clear that the policy-side improvement is **reproducible rather than seed-dependent**.
>
> ---
>
> ### Q3: Sensitivity to $m$ and $P_{\min}$.
>
> **Response:**
>
> **Sensitivity to $m$.** We ran **$m \in \{0, 5, 10\}$** on **Qwen3-0.6B** and report both **BoN** accuracy and **PB** F1:
>
> | $m$ | BON-AVG | PB-AVG |
> |---|---:|---:|
> | 0 | 26.1 | 37.0 |
> | **5** | **28.1** | **39.1** |
> | 10 | 27.5 | 39.1 |
>
> These results show that the method is **stable around the default choice $m=5$**. Moving from $m=5$ to $m=10$ changes performance only slightly, while removing the margin ($m=0$) causes a clearer degradation, especially on BoN.
>
> **Sensitivity to $P_{\min}$.** We sampled **1,000 training prompts**, rolled out the **Qwen3-0.6B SFT policy**, and recorded full-vocabulary next-token probabilities at each step. From these distributions, we computed the **average candidate set size** and **retained policy probability mass**:
>
> | $P_{\min}$ | Avg Candidate Size | Prob Mass Coverage |
> |---|---:|---:|
> | 0.05 | 1.54 | 97.57% |
> | **0.10** | **1.33** | **96.11%** |
> | 0.20 | 1.16 | 93.58% |
>
> These results make $P_{\min}=0.1$ a well-justified operating point: it keeps the candidate set very small in practice while still retaining most of the policy mass. Lower thresholds yield only limited additional coverage, whereas larger thresholds start to discard nontrivial policy mass.
>
> ---
>
> ### Q4: Math-only evaluation / scope and limitations.
>
> **Response:**
> We agree that our evaluation is currently limited to math-like tasks with deterministic verifiers, and we will state this explicitly in the revision. We chose this setting because it enables cleaner evaluation with less subjective and noisy judgment. Our claims are therefore limited to this verifiable reasoning setting.

---

> > ### Author Rebuttal · Reviewer_4Qnz · 2026-04-06
> >
> > I thank the authors for the detailed rebuttal, especially the policy-side ablation disentangling IPVRM from auxiliary components, and the multi-seed statistics for DistRL. As my original score was already a weak accept, I am inclined to maintain my current score.

---

### Official Review · Reviewer_QMYr · 2026-03-16

**Soundness:** 2
**Presentation:** 2
**Significance:** 2
**Originality:** 2
**Overall Recommendation:** 4
**Confidence:** 2

**Summary:**

This paper argues that existing implicit process reward models suffer from a fundamental train–inference mismatch: they are trained only through a sequence-level constraint, yet are used at inference time as if they provided faithful token-level assessments of step correctness. To address this, the authors propose IPVRM, an implicit prefix-value reward model that directly estimates the correctness probability of a partial reasoning prefix, and derives stepwise supervision through temporal-difference differences rather than post hoc reward decomposition. Building on this, they introduce DistRL, a distribution-level reinforcement learning objective that leverages reward estimates over candidate next tokens, enabling denser and more sample-efficient policy optimization. Across reward-model evaluation and downstream RL experiments, the paper’s main claim is that better prefix-value estimation leads to more reliable process supervision and stronger reasoning performance than prior implicit reward formulations.

**Compliance With Llm Reviewing Policy:**

Affirmed.

**Key Questions For Authors:**

Can you disentangle how much of the improvement comes from the core IPVRM formulation itself, versus the additional design and stabilization choices around it?

**Limitations:**

yes

**Strengths And Weaknesses:**

Strengths
1. The paper identifies a clear and important weakness in prior implicit PRMs, namely the train–inference mismatch between sequence-level training and token-level use. The proposed IPVRM is conceptually clean and well motivated, since it directly learns prefix-conditioned values instead of relying on ambiguous reward decomposition.
2. The approach retains the practical benefits of implicit reward modeling, including low annotation cost and easy online updating during RL.
3. The experimental narrative is consistent and supports the claim that better prefix-value estimation leads to stronger process supervision.

Weaknesses
1. The method bundles many auxiliary choices, so the gain is not cleanly attributable to the core IPVRM idea.
2. The advantage of the proposed method over generative PRMs is exaggerated as in practice generative prm overhead is acceptable.

---

> ### Author Rebuttal · Authors · 2026-03-29
>
> # Response to Reviewer QMYr
>
> Thank you for the thoughtful feedback and for raising an important attribution question.
>
> ### Q1: The method bundles many auxiliary choices, so the gain is not cleanly attributable to the core IPVRM idea. Can you disentangle how much of the improvement comes from the core IPVRM formulation itself, versus the additional design and stabilization choices around it?
>
> **Response:**
> Yes. We address this directly with two additional analyses—**RM-only** and **policy-side ablation**—which separate the contribution of the core IPVRM reformulation from the contribution of later stabilization components. The main conclusion is that IPVRM itself already delivers substantial gains, while the later components provide further improvements.
>
> **(a) RM-only evidence.** Before any RL-stage components such as DistRL, ADB, or DLW are introduced, replacing Implicit PRM with IPVRM already yields clear improvements in reward-model evaluation. On **Qwen3-0.6B** (from the main paper):
> - **Best-of-N average** improves from **25.1** (Implicit PRM) to **28.1** (IPVRM).
> - **ProcessBench average F1** improves from **25.4** (Implicit PRM) to **39.1** (IPVRM) under the protocol used in the main paper.
>
> Because these gains appear **before** DistRL, ADB, or DLW are introduced, they directly isolate the benefit of the **core prefix-value formulation itself**.
>
> **(b) Policy-side ablation.** We also conducted an additional controlled policy-side ablation on **Qwen3-0.6B**:
>
> | Method | AIME | MATH-500 | MinervaMath | Olympiad | AMC23 | AVG |
> |---|---:|---:|---:|---:|---:|---:|
> | DistRL + Implicit PRM w/o ADB/DLW | 0.8 | 37.1 | 10.9 | 10.5 | 14.2 | 14.70 |
> | DistRL + IPVRM w/o ADB/DLW | 1.0 | 40.8 | 11.2 | 12.2 | 12.0 | 15.44 |
> | DistRL + IPVRM | 1.2 | 40.8 | 13.1 | 12.3 | 13.5 | 16.18 |
>
> These results show that:
>
> - Replacing **Implicit PRM → IPVRM** within the same DistRL pipeline already improves AVG from **14.70 → 15.44** (**+0.74**).
> - Adding **ADB/DLW** on top of IPVRM further improves AVG from **15.44 → 16.18** (**+0.74**).
> These results make the attribution much clearer: the gains are **not** driven solely by stabilization tricks. The **core IPVRM formulation itself** already produces a substantial improvement, and ADB/DLW contribute an additional gain.
>
> ---
>
> ### Q2: The advantage of the proposed method over generative PRMs is exaggerated as in practice generative prm overhead is acceptable.
>
> **Response:**
> This is a fair point. We will state this comparison in the revision to be more precise. That said, our primary focus is on online RL settings that require frequent reward-model refreshes, where the inference latency and annotation requirements of generative PRMs become a significant bottleneck.

---

> > ### Author Rebuttal · Reviewer_QMYr · 2026-04-02
> >
> > I keep my rating at 4.

---

### Decision · Program_Chairs · 2026-04-30

**Decision:**

Accept (regular)

**Comment:**

This paper proposes Implicit Prefix-Value Reward Model (IPVRM), a new method to learn a prefix-conditioned value function estimating the probability of eventual correctness, and derives step signals via temporal-difference (TD) differences. The motivation is to mitigate the noise of existing PRM training methods that learns from trajectory-level outcomes and only constrains a sequence-level aggregate. This is an important direction as PRMs can potentially significant improve the efficiency of RL training while existing PRMs are barely useful. With the proposed approach, IPVRM improves on the ProcessBench by a large margin, but only achieves limited gains when applied to RL training. The first round of the reviews questioned the source of the gain given multiple components in the method, as well as statistical significance for some of the results. The authors did a good job in the rebuttal to address these concerns and all the reviewers selected "fully solved" in the rebuttal ack. All the reviewers gave a score of 4 as a weak accept, and I would vote for weak accept as well. I am not recommending higher mainly because the proposed PRM does not perform well in RL experiments.